# Extracting and Transferring Abilities For Building Multi-lingual Ability-enhanced Large Language Models

## Abstract

Multi-lingual ability transfer has become increasingly important for the broad application of large language models (LLMs). Existing work highly relies on training with the multi-lingual ability-related data, which may be not available for low-resource languages. To solve it, we propose a **M**ulti-lingual **A**bility **E**xtraction and **T**ransfer approach, named as **MAET**. Our key idea is to decompose and extract language-agnostic ability-related weights from LLMs, and transfer them across different languages by simple addition and subtraction operations without training. Specially, our MAET consists of the extraction and transfer stages. In the extraction stage, we firstly locate key neurons that are highly related to specific abilities, and then employ them to extract the transferable ability-specific weights. In the transfer stage, we further select the ability-related parameter tensors, and design the merging strategy based on the linguistic and ability specific weights, to build the multi-lingual ability-enhanced LLM. To demonstrate the effectiveness of our proposed approach, we conduct extensive experiments on mathematical and scientific tasks in both high-resource lingual and low-resource lingual scenarios. Experiment results have shown that MAET can effectively and efficiently extract and transfer the advanced abilities, and outperform training-based baselines methods. Our code and data will be publicly released.

## 1 Introduction

Large language models (LLMs) have recently shown remarkable performance on various general tasks, *e.g.,* text generation and question answering (Zhao et al., 2023; OpenAI, 2023; Dubey et al., 2024). Despite the success, LLMs are still struggling to solve complex tasks (*e.g.,* mathematical reasoning), which require LLMs to possess specific advanced abilities (*e.g.,* deductive reasoning) and knowledge (*e.g.,* mathematical theory) (Yue et al., 2024; Lu et al., 2022). To address it and further improve LLMs, existing work either collects the related data to train LLMs (Du et al., 2024; Chen et al., 2024a), or merges the parameters of existing well-performed LLMs to transfer their advanced abilities into one single model (Ilharco et al., 2023; Yadav et al., 2023; Yu et al., 2024a).

Despite the success, it is not easy to collect sufficient training corpus or well-trained LLMs related to specific abilities, especially in multi-lingual scenarios. Especially, some popular languages (*e.g.,* English) have dominated the linguistic expressions of the open web data, and the amount of available domain-specific data for low-resource languages (*e.g.,* Bengali or Telugu) is highly limited (Maguer-esse et al., 2020; Patzelt, 2024; Mirashi et al., 2024). Fortunately, existing work (Zhao et al., 2024; Schäfer et al., 2024) has revealed that the learned knowledge from one language by LLMs could be inherited and leveraged by other languages. For example, Llama-series LLMs are trained mainly on English texts, while they can also solve the tasks based on other languages. Such a finding has been widely explored in either improving the overall performance of multi-lingual LLMs (Schäfer et al., 2024) or enhancing fine-grained knowledge (Chen et al., 2024a). However, the related work mostly relies on training with ability-related corpus in the target language, which is not always available for low-resource languages.

To conduct a more effective ability transfer, our idea is to learn and extract the "*ability-specific weights*" that preserves the knowledge about specific abilities for the LLM. If such ability-specific

and language-specific weights could be decomposed, it is achievable to transfer the required abilities into target languages by just combining the corresponding weights, even building a multi-lingual ability-enhanced LLM like building blocks. Based on this idea, in this paper, we propose a **M**ulti-lingual **A**bility **E**xtraction and **T**ransfer approach, named as **MAET**. Specifically, our approach consists of two major stages, *i.e.*, ability extraction and transfer stage. In the extraction stage, we first locate the abilities-related neurons and leverage related corpus in a reference language to continually pre-train the LLM on these identified neurons. Then, based on the LLM trained on the general corpus, we devise the formula to extract the ability-specific weights. In the transfer stage, we utilize the ability-related weights to select related parameter tensors, and design a specific model merging strategy by interpolating linguistic and ability-specific weights. In our approach, we only need ability-specific corpus from any rich-resource language and general multi-lingual corpus, which can effectively mitigate the data scarcity issues in low-resource languages.

To assess the effectiveness of our approach, we conduct the evaluation based on two comprehensive reasoning benchmarks, namely Multi-lingual Grade School Math (MGSM) (Shi et al., 2023) and science tasks from multi-lingual MMLU (Lai et al., 2023) as the evaluation benchmarks. According to the evaluation results, the proposed approach MAET outperforms other competitive baseline methods (*e.g.,* continual pre-training (Gururangan et al., 2020) and model merging methods with task vectors (Ilharco et al., 2023), achieving the 9.1% relative improvement compared to the base LLM. In addition, our approach can work well with relatively fewer training data, demonstrating an improved efficiency in practice. In conclusion, our contribution can be summarized as follows,

(1) Our research has found that advanced abilities can be extracted from the single-lingual corpus and transferred across languages without the multi-lingual ability-related corpus, enabling to efficiently empower LLMs with special advanced abilities.

(2) We propose an effective and efficient approach named MAET, which first identifies and extracts the ability-related weights in LLMs and then only leverages simple addition and subtraction operations to build a multi-lingual ability-enhanced LLM.

(3) Our approach MAET achieves better performance than the competitive baseline methods (*e.g.,* continual pre-training and model merging with task vector) in multi-lingual complex reasoning tasks, including mathematical reasoning tasks and scientific reasoning tasks.

## 2 RELATED WORK

We introduce the related work from the following three perspectives:

**Continual Pre-training.** Although LLMs have shown remarkable performance on various downstream work, they still struggle in several specific tasks, *e.g.,* complex reasoning tasks (Paster et al., 2024; Shao et al., 2024) or low-resource lingual scenarios (Hedderich et al., 2021; Panchbhai & Pankanti, 2021). To adapt LLMs pre-trained on the general corpus to multi-lingual scenarios or specific tasks, existing work (Luo et al., 2022; Taylor et al., 2022; Zhao et al., 2022; Zhang et al., 2024a) has collected the corresponding corpus to continually pre-train (CPT) LLMs. During the continual pre-training process, the mixture strategy between the general corpus and the ability-related corpus should be carefully considered to prevent hurting the general abilities of LLMs (Ye et al., 2024; Xie et al., 2023; Chen et al., 2024a; Siriwardhana et al., 2024). However, previous study Chang et al. (2024); Lu et al. (2023) has pointed out that it is difficult to collect the required corpus, especially for low-resource language corpus. Therefore, synthesizing data from powerful LLMs is widely utilized to expand the task-specific training corpus (Chen et al., 2021b; Yu et al., 2024b; Zhou et al., 2024a). Besides, because of the limitation of computation resources, a series of approaches (Hu et al., 2022; Li & Liang, 2021; Dettmers et al., 2023) only train several parameters to reduce the expenses. In this work, we focus on adapting LLMs to multilingual complex reasoning scenarios through continually pre-training LLMs on the single-lingual task-specific corpus.

**Knowledge Editing.** According to lottery ticket hypothesis (Frankle & Carbin, 2019), training a small number of model parameters will achieve comparable or even better performance on downstream tasks. Existing study (Du et al., 2024; Wang et al., 2024b; Gong et al., 2024) has leveraged the inner information of LLMs, *e.g.,* gradient or cosine similarity between different hidden states, to select and train the related sub-network. Besides, the probe (*i.e.,* a newly initialized parameter) can

be implemented to detect the knowledge of LLMs and process targeted repair (Wang et al., 2024a; Jiang et al., 2024). However, Since these approaches need to calculate and select a sub-network of each training instance, which might cause the instability of the training process, several study (Chen et al., 2024b; Zhang et al., 2024b) pointed out that the task-related sub-network can be determined before the training process, and only updating the value of the corresponding neurons can achieve better performance. In this work, we focus on editing the task-specific neurons of LLMs to improve the corresponding capacities in multi-lingual scenarios.

**Model Merging.** Given that the CPT process will bring huge computational expenses, previous work leveraged model merging techniques to integrate different abilities (*e.g.,* mathematical reasoning and code synthesizing) into one model (Yang et al., 2024; Xu et al., 2024b; Stoica et al., 2024). During the merging process, the interference between different LLMs might be conflict with each other and affect the final performance. Therefore, researchers proposed the clip (Yadav et al., 2023) or randomly dropout (Yu et al., 2024a) mechanism to mitigate the performance decrease. Moreover, the selection of the hyper-parameters (*e.g.,* weight of each model) is the challenge of the model merging process, and previous work (Zhou et al., 2024b; Matena & Raffel, 2022) utilized the inner parameters of LLMs or external matrixes to determine the hyper-parameters. Furthermore, a series of work has studied improving the reasoning ability of LLMs in non-English scenarios by merging the reasoning-specialized model and multi-lingual model (Huang et al., 2024; Yoon et al., 2024). Inspired by the above work, we try to locate the task-related sub-networks of LLMs and transfer the advanced abilities.

## 3 PRELIMINARY

Despite that LLMs exhibit excellent performance on general tasks, they still have limited advanced abilities, *e.g.,* mathematical and scientific reasoning abilities. A typical approach to enhancing these abilities is to continually pre-train (CPT) LLMs with ability-related corpus. However, such training data might not always be available or sufficient, especially for minor domains (*e.g.,* Bengali). In this work, we focus on the task of *ability extraction and transfer* by continual pre-training and merging LLMs. Concretely, LLMs are trained on collected corpus from a certain domain, and we aim to only transfer its learned advanced capabilities to another target domain (Zhuang et al., 2021; Farahani et al., 2021) without further training efforts. In this work, we mainly study the cross-lingual scene where the linguistic-agnostic advanced ability and linguistic abilities should be extracted and transferred, to help build a unified multi-lingual ability-enhanced LLM.

Formally, for a certain ability $A_i$ and a set of languages $L = \{L_0, L_1, \ldots, L_n\}$, we assume that the general corpus of all languages can be collected, denoted as $C_{\text{general}} = \{C_{L_0}, C_{L_1}, \ldots, C_{L_n}\}$, while the ability-related corpus is only available in language $L_0$ (*i.e.,* English), denoted as $C_{L_0, A_i}$. Based on the above corpora, our goal is to extract and transfer the advanced ability $A_i$ from language $L_0$ and linguistic abilities from other languages $L_1, \ldots, L_n$, into a unified LLM.

## 4 APPROACH

In this section, we propose the **M**ulti-lingual **A**bility **E**xtraction and **T**ransfer approach, named as **MAET**, which can effectively transfer the advanced abilities from single-lingual LLMs, to build a multi-lingual ability-enhanced LLM. The key motivation of our approach is to identify and extract ability-related neurons or weights, and transfer the target abilities into a LLM in an efficient way. The overall framework of MAET is presented in Figure 1.

### 4.1 ABILITY-RELATED WEIGHTS EXTRACTION

In this part, we aim to locate and learn ability-related parameter weights within an LLM, to enable efficient transferring of the ability into other LLMs. Concretely, it consists of two major steps, *i.e.,* key neurons locating and ability-related parameter weights learning, which are detailed in the following.

**Locating the Key Neurons.** The gradient of each neuron in LLMs can be utilized to estimate its correlation degree with specific task ability , we select those with high gradient values as key neurons.

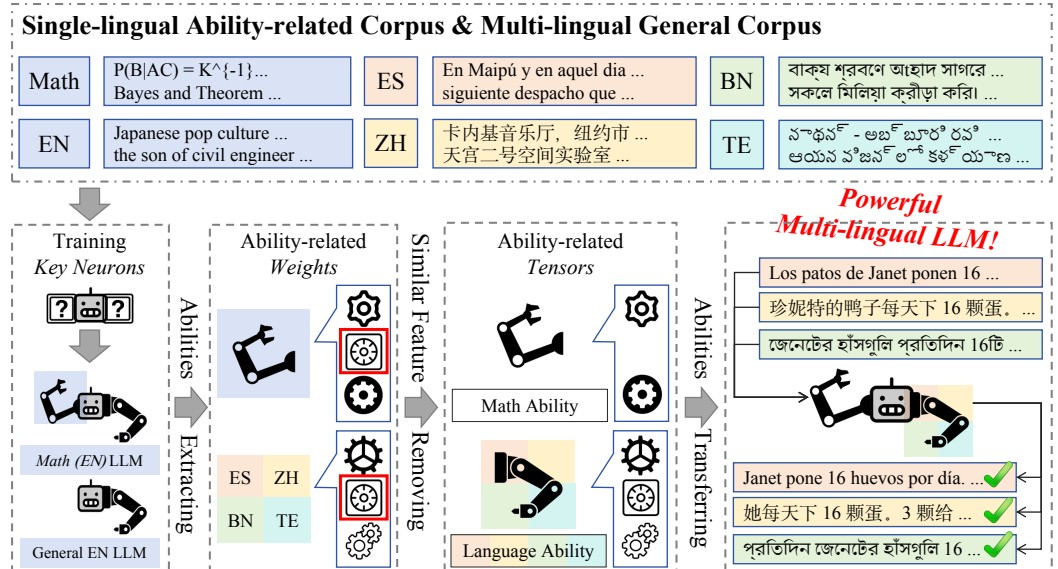

Figure 1: The framework of our approach MAET, including extraction and transfer stages. In the extraction stage, we first locate the key neurons, and utilize the single-lingual ability-related corpus and general corpus to train the LLM on these neurons to obtain the ability-related weight. Then, we remove the parameter tensors related to language knowledge in the ability weight and transfer the remaining to the base LLM. After these stages, we can obtain a powerful LLM with advanced abilities that can solve the corresponding tasks in multi-lingual scenarios.

To this end, we first use the ability-related corpus $C'_{L_0, A_i}$ to continually pre-train the LLM, while sampling a small amount to train the model can be also applied to reduce the computation consumption. During training, the LLM learns the language modeling task and each neuron is updated by the gradients associated by the training instances. Due to the high cost of calculating the accumulation of gradient at each training step, we calculate the value changes of the LLM neurons before and after the training process to approximate the importance. Formally, the importance function $I(A_i, \theta_j)$ of neurons can be computed as:

$$I(A_i, \theta_j) = \sum_{d_k \in C'_{L_0, A_i}} \text{Gradient}\left(\theta_j, d_k\right) \approx \lambda \cdot \parallel \tilde{\theta}_j - \theta_j \parallel, \qquad (1)$$

where $d_k$ denotes the $k$-th instance of training corpus $C'_{L_0, A_i}$ and $\tilde{\theta}_j$ denote the value of the $j$-th neuron of LLM after training, respectively. Based on it and inspired by previous work (Yadav et al., 2023), we rank all neurons according to their importance scores, and then select the top $k_1$% ones into the set $\mathcal{N}_{A_i}$ as the key neurons.

**Learning Ability-related Weights.** Based on the identified key neurons in $\mathcal{N}_{A_i}$, we further learn the ability-related parameter weights. Our motivation is to decompose the parameter weights according to their changes *before* and *after* the LLM has mastered a specific ability, which is achievable owing to the modularity and composition nature of the LLM parameter matrices (Yu et al., 2024a; Shazeer et al., 2017). First, we utilize the key neurons locating method mentioned above to extract the ability-related neuron set $\mathcal{N}_{A_i}$, and also obtain the language-related neuron set $\mathcal{N}_{L_0}$ via the same way. Then, we train the LLM with the mixture of ability-related corpus and general corpus on the key neuron set $\mathcal{N}_{A_i} \bigcup \mathcal{N}_{L_0}$ and $\mathcal{N}_{L_0}$ respectively, to obtain two specific models, denoted as $\text{LLM}_{A_i, L_0}$ with parameters $\Theta_{A_i, L_0}$ and $\text{LLM}_{L_0}$ with parameters $\Theta_{L_0}$. Next, we measure the parameter changes between the backbone and the trained models, and obtain the ability-related weights via the parameter decomposition operation as:

$$R(A_i) = \alpha \cdot \underbrace{(\Theta_{A_i, L_0} - \Theta_o)}_{\text{Ability \& language difference}} - \beta \cdot \underbrace{(\Theta_{L_0} - \Theta_o)}_{\text{Language difference}}, \qquad (2)$$

where $\alpha$ and $\beta$ are tunable coefficients to balance the two parts of weight differences, and $\Theta_o$ denote the original parameters of the LLM, which serves as the reference for parameter decomposition. As we only train the parameters within the neuron set, its weight difference should preserve the knowledge about the corresponding ability. Thus, it can be regarded as the *ability-related parameter representations*, and is promising to transfer the ability into other LLMs by the addition operation.

## 4.2 MULTI-LINGUAL ABILITY TRANSFER

After obtaining the ability-related weights, in this part, we utilize them to transfer and integrate the abilities, to build a multi-lingual ability-enhanced LLM.

**Ability-related Parameter Tensor Selection.** Although we can locate the ability-related key neurons, it is still hard to avoid the involvement of irrelevant ones. Our empirical studies have found that neuron-level features are easy to be affected by the noisy data. Therefore, we consider identifying ability-related parameter tensors, which correspond to the parameter matrices within the LLM. Specifically, we firstly leverage the ability-related weights of languages $R(L_1), \ldots, R(L_n)$ to obtain the multi-lingual weight $R_{Lang}$. Given that large models have varying levels of proficiency in different languages, we use the hyper-parameters $\mu_1, \ldots, \mu_n$ to tune this process as:

$$R_{Lang} = \sum_{i=1}^{n} \mu_i \cdot R(L_i), \tag{3}$$

where $R(L_i)$ preserves the linguistic ability of language $L_i$ learned based on Equation 2. Therefore, $R_{Lang}$ can be considered as the general language ability of LLMs that spans multiple languages. As we aim to find he parameter tensors that have low linguistic effects but focus on the desired abilities (*e.g.,* mathematical reasoning), we rank all the tensors according to their similarities with $R_{Lang}$, and pick up the last $k_2\%$ ones. Formally, for tensor $\tau_i$, we calculate the cosine similarity of this parameter between $R(A_i)$ and $R_{Lang}$, as follows,

$$S(\tau_i) = \text{sim}\left(R(A_i)[\tau_i], R_{Lang}[\tau_i]\right) = \frac{R(A_i)[\tau_i] \cdot R_{Lang}[\tau_i]}{|R(A_i)[\tau_i]| \times |R_{Lang}[\tau_i]|}, \tag{4}$$

where we use the cosine similarity to implement the similarity function $\text{sim}(\cdot)$. After obtaining the similarity of all tensors, we rank them in a descending order based on the similarity values, and then select the last $k_2\%$ parameters into the set $\mathcal{T}$ as the ability-related parameters.

**Building Multi-lingual Ability-enhanced LLM.** Based on the selected ability-related tensors $\mathcal{T}$, we design the model merging process by interpolating ability weights and multi-lingual weights, to build the multi-lingual ability-enhanced LLM. Formally, the final parameter tensors of the target LLM are computed as:

$$\tilde{\tau}_i = \tau_i^{(o)} + \begin{cases} \gamma \cdot R(A_i)[\tau_i] + \eta \cdot R_{Lang}[\tau_i], & \tau_i \in \mathcal{T} \\ R_{Lang}[\tau_i], & \tau_i \notin \mathcal{T} \end{cases}, \tag{5}$$

where $\tau_i^{(o)}$ denotes the original value of parameter tensor $\tau_i$, and $\gamma$ and $\eta$ are tunable hyper-parameters. This formula can be explained in two different cases. When a parameter tensor serves as the major role for specific abilities, we update it by adding both ability- and linguistic-related weights; otherwise, we simply enhance it with multi-lingual weights. In this way, we can derive a more powerful LLM that is equipped with the multi-lingual abilities and specific advanced abilities.

## 4.3 THE OVERALL PROCEDURE

To better demonstrate our approach, we present key concepts in Table 5 for further clarifying and provide the complete procedure in Algorithm 1 in the pseudo-code form. The procedure of MAET consists of two main stages, *i.e.,* ability-related weights extraction and multi-lingual ability transferring. For the extraction stage, we first utilize the accumulated gradient to estimate the importance of each neuron by Equation 1. Then, we leverage the model trained on the general corpus to remove the influence of language and obtain the ability-related weight through Equation 2. In the transfer stage, we utilize Equation 3 and Equation 4 to obtain the multi-lingual weight and identify the

| Aspects | CPT | MoE | LoRA | MoL | TV | MAET (Ours) |
|---|---|---|---|---|---|---|
| MLAR Corpus | Yes | Yes | Yes | Yes | Yes | **No** |
| Tuning Parameters | Full | Full | Low-Rank | Low-Rank | Full | **Ability-related** |
| Ability Composition | No | No | No | No | Yes | **Yes** |
| Ability Transfer | No | No | No | No | No | **Yes** |

Table 1: The difference between our proposed MAET and the methods in previous work (*i.e.,* CPT (Hu et al., 2022), Mixture-of-Expert (MoE) (Shazeer et al., 2017), LoRA (Hu et al., 2022), Mixture-of-LoRA (MoL) (Feng et al., 2024), and Task Vector (TV) (Ilharco et al., 2023). MLAR denotes the abbreviation of multi-lingual ability-related corpus.

---

**Algorithm 1:** The complete procedure of our proposed approach.

---

**Input** : Single-lingual ability-related corpus $C_{L_0, A_i}$, multi-lingual general corpus $C_{L_0}, C_{L_1}, \ldots, C_{L_n}$, and the parameters of the backbone model $\Theta_o$.

**Output:** A well-trained multi-lingual ability-enhanced LLM.

```
// Ability-related Weights Extraction
```
1  $\theta' \leftarrow \text{CPT}(C_{L_0, A_i}, \Theta_o)$;
2  **for** *j-th neuron in* $\Theta_o$ **do**
3  $\quad$ Calculate the importance score of the corresponding neuron using Eq. 1;
4  Identify the key neuron set $\mathcal{N}_{A_i}$;
5  $\text{LLM}_{A_i, L_0} \leftarrow \text{CPT}(C_{L_0, A}, \Theta_o, \mathcal{N}_{A_i} \cup \mathcal{N}_{L_0})$;
6  $\text{LLM}_{L_0} \leftarrow \text{CPT}(C_{L_0}, \Theta_o, \mathcal{N}_{L_0})$;
7  Learning the ability-related weight $R(A_i)$ using Eq. 2;
```
// Multi-lingual Ability Transfer
```
8  Obtaining the multi-lingual weight $R_{Lang}$ using Eq. 3;
9  **for** *j-th parameter tensor in LLM* **do**
10 $\quad$ Calculate the correlation using Eq. 4;
11 Identify the ability-related parameters $\mathcal{T}$;
12 Transfer the ability to multi-lingual scenarios using Eq. 5;

13 Obtain the well-trained multi-lingual ability-enhanced LLM.

---

ability-related parameter tensors in LLM. After it, we leverage Equation 5 to fulfill the multi-lingual ability transfer, to build the multi-lingual ability-enhanced LLM.

To highlight the difference between our approach and previous work, we present the comparison of these methods in Table 1. To adapt LLMs to multi-lingual scenarios, most of the existing methods (*e.g.,* CPT and TV) require the multi-lingual ability-related corpus (*i.e.,* ability-related corpus is required for each language) for training the LLM parameters. In comparison, our proposed approach only trains and modifies the ability-related parameters, which can efficiently focus on enhancing the specific ability. A major novelty of our work is that we identify the key units and implement the sparse update in the model training and merging procedure, which can effectively decompose, extract, and transfer the abilities of LLMs. In addition, compared with the LoRA-based methods (*i.e.,* LoRA and MoL) that also sparsely update the LLM parameters, our approach selectively updates the ability-related neurons, while LoRA-based methods utilize the low-rank matrices to approximate the original parameters.

## 5 EXPERIMENT

### 5.1 EXPERIMENTAL SETTINGS

In this part, we introduce the details of our experimental settings, including the datasets utilized in the continual pre-training and evaluation process, baseline methods for comparison, and the implementation details of our approach.

**Datasets.** In this work, we focus on transferring the advanced abilities (*i.e.,* mathematical and scientific reasoning abilities) of LLMs from English scenarios to multi-lingual scenarios, including high-resource languages (*i.e.,* Spanish and Chinese) and low-resource languages (*i.e.,* Bengali and Telugu). For the training corpus, we extract the corpus of the corresponding languages from the dataset proposed by previous work (Yang et al., 2023; Scao et al., 2022; Laurençon et al., 2022) as the general training corpus, and utilize OpenWebMath (Paster et al., 2024) and the arXiv papers (Soldaini et al., 2024) as the ability-related corpus for mathematical tasks and scientific tasks respectively. For the evaluation benchmark, we follow the evaluation settings in previous work (OpenAI, 2023), utilizing *Multi-lingual Grade School Math (MGSM)* (Shi et al., 2023) and science tasks from *multi-lingual MMLU* (Lai et al., 2023) (*i.e.,* college biology, college chemistry, college physics, high school biology, high school chemistry, and high school physics) as the downstream tasks for multi-lingual scenarios. The statistical information of the datasets is presented in Table 7.

**Baselines.** In our evaluation, a baseline can be represented as three parts, *i.e.,* training parameters, training approach, and training data. First, we conduct the full parameters training and the LoRA training (Hu et al., 2022) in our evaluation, denoted as *the "F" and "L" at the prefix* of the training approaches, respectively. For the training approach, we employ *continual pre-training (CPT)* (Gururangan et al., 2020), *domain adaption (DA)* (Taylor et al., 2022), and *model merging with task vector (TV)* (Ilharco et al., 2023). Besides, for the training data, *"L", "A", and "T"* refer to the multi-lingual general corpus, the single-lingual ability-related corpus, and the translated multi-lingual ability-related corpus from GPT-4o (Hurst et al., 2024), respectively. Moreover, to conduct a more comprehensive evaluation, we also present the performance of different LLMs, *i.e.,* Baichuan-2 7B (Yang et al., 2023), Mistral 7B (Jiang et al., 2023), LLaMA-2 7B (Touvron et al., 2023), and LLaMA-3 8B (Dubey et al., 2024).

**Implementation Details.** In the experiment, we adapt LLaMA-3 8B as the backbone LLM, and employ `Transformers` (Wolf et al., 2020) and `Deepspeed` framework to perform the training process. And we also present the evaluation results of different backbone LLM (*i.e.,* Qwen2.5 0.5B (Hui et al., 2024) and Gemma2 2B (Rivière et al., 2024)) in Appendix E. For the training process, the learning rate, batch size, and training step are set as $5 \times 10^{-5}$, 1M tokens, and 2B tokens, respectively. Besides, for the key neurons locating, we select the top 5% relevant neurons as the key neuron set $\mathcal{N}$ for both stages and identify the last 80% and 60% similar tensor as the key sub-network $\mathcal{T}$ for mathematical reasoning tasks and scientific reasoning tasks respectively. We present model details about hyper-parameters in Appendix B.

## 5.2 MAIN RESULTS

To comprehensively evaluate our proposed MAET, we employ MAET on mathematical and scientific tasks in multi-lingual scenarios and present the results in Table 2.

First, MAET outperforms other baseline methods in the average performance of all downstream languages, and even achieves better performance than CPT-based methods (*i.e.,* F-CPT$_{L\&A}$ and L-CPT$_{L\&A}$), which consuming double computation resource than our approach. The experiment results show that MAET maintains the balance of advanced abilities of LLMs on different linguistic tasks and improves the backbone LLM advanced abilities on multi-lingual scenarios effectively and stably. Without the multi-lingual ability-related training corpus, MAET can extract the ability weights from the single-lingual corpus and transfer the abilities of multi-lingual scenarios, while other methods cannot attain the abilities transfer.

Second, continual pre-training LLMs on the mixture of multi-lingual general corpus and single-lingual ability-related corpus (*i.e.,* F-CPT$_{L\&A}$) can enhance the specific ability of LLMs, achieving the second best performance. However, when adapting LLMs to a new domain or enhancing a new ability of LLM, CPT-based methods should retrain the LLMs on the ability-related and multi-lingual corpus, showing that CPT is leaked of transferability and requires more computational resources. For the issue of new domain adapting, MAET only utilizes a small amount of single-lingual ability-related corpus (*i.e.,* English corpus in practice) to obtain the ability weight, which can be employed to transfer the corresponding advanced ability, achieving both effectiveness and efficiency.

Third, LoRA-based methods (Hu et al., 2022) (*e.g.,* L-CPT$_{L\&A}$, L-CPT$_L$, L-TV) initialize the low-rank matrices and only update these matrices, performing sparsely optimize on LLM. Since the

| Methods | Multilingual Mathematical Tasks | | | | | Multilingual Scientific Tasks | | | | |
|---|---|---|---|---|---|---|---|---|---|---|
| | ES | ZH | BN | TE | Avg. | ES | ZH | BN | TE | Avg. |
| Baichuan-2 7B | 17.20 | 28.00 | 4.80 | 2.40 | 13.10 | 42.27 | 46.43 | 30.17 | 26.21 | 36.27 |
| Mistral 7B | 38.80 | 34.40 | 9.60 | 2.80 | 21.40 | 52.08 | 45.33 | 32.91 | 27.96 | 39.57 |
| LLaMA-2 7B | 7.60 | 12.00 | 1.60 | 0.00 | 5.30 | 34.16 | 31.68 | 24.56 | 22.15 | 28.14 |
| LLaMA-3 8B | 48.40 | 38.80 | 28.80 | 20.40 | 34.10 | 55.06 | 47.24 | 36.63 | 29.26 | 42.05 |
| + F-CPT$_{L\&A}$ | 46.80 | **42.00** | 28.40 | **27.60** | 36.20 | 55.92 | 48.57 | 36.84 | 30.10 | 42.86 |
| + L-CPT$_{L\&A}$ | 44.80 | 37.60 | 28.80 | 23.60 | 33.70 | 54.77 | 46.81 | 36.41 | 29.88 | 41.97 |
| + F-CPT$_{A\&T}$ | - | - | - | - | - | 53.73 | 46.30 | 35.06 | 31.73 | 41.71 |
| + F-CPT$_A$ | - | - | - | - | - | 51.90 | 45.71 | 33.35 | 29.41 | 40.09 |
| + F-CPT$_T$ | - | - | - | - | - | 50.35 | 45.36 | 34.54 | **34.46** | 41.18 |
| + F-CPT$_L$ | 38.80 | 35.60 | 28.00 | 23.60 | 31.50 | 53.56 | 47.14 | 35.89 | 30.64 | 41.81 |
| + F-CPT$_L$ & DA | 41.60 | 39.60 | **34.40** | **27.60** | 35.80 | 52.71 | 48.05 | 35.49 | 28.62 | 41.11 |
| + L-CPT$_L$ | 46.40 | 39.20 | 28.40 | 22.80 | 34.20 | 55.04 | 48.09 | 36.66 | 30.43 | 42.56 |
| + L-CPT$_L$ & DA | 46.80 | 37.60 | 28.00 | 27.20 | 34.90 | 55.65 | **49.10** | 36.48 | 29.65 | 42.72 |
| + F-TV | 42.00 | 32.40 | 16.00 | 10.40 | 25.20 | 53.36 | 46.57 | 36.70 | 30.73 | 41.84 |
| + L-TV | 45.60 | 39.20 | 30.80 | 25.60 | 35.30 | 55.46 | 48.27 | 36.65 | 30.44 | 42.71 |
| + MAET (Ours) | **49.60** | 41.60 | 32.40 | 25.20 | **37.20** | **56.20** | 48.00 | **37.64** | 30.38 | **43.06** |

Table 2: The performance comparison of different approaches on multilingual mathematical tasks and multilingual scientific tasks. Avg. denotes the average accuracy of the multi-lingual tasks. ES, ZH, BN, and TE denote Spanish, Chinese, Bengali, and Telugu, respectively. The best is in bold and the second best is underlined.

trainable parameters in LoRA represent the whole parameters of LLM rather than ability-related sub-network, it cannot perform well on the multi-lingual scenarios, indicating the failure of the advanced abilities transferring. In contrast, MAET first identifies the ability-related sub-networks and utilizes the corresponding sub-networks to perform the following operations. Because of the decomposing of the inner abilities of LLMs, MAET can help LLMs improve their specific ability.

Fourth, translation-based methods are the strong baselines to enhance the LLM performance in low-resource languages. In the experiment, we utilize GPT-4o to translate the ability-related corpus from English to other languages, and present the prompt in Appendix D. According to the experimental results in the above table, we can observe that our MAET outperforms the translation-based method. The translation-based method consumes more computational resources and cannot achieve better performance. The reason might be that the translated corpus shares similar knowledge of the specific domain, which makes LLM overfit the corresponding knowledge and cannot really understand the specific knowledge. In contrast, our approach decomposes the scientific ability and language ability, and transfers the scientific ability from one language to another, preventing overfitting, decreasing the expense, and improving performance.

Last, compared with the model merging based approaches (*i.e.,* F-TV and L-TV), experimental results have shown that MAET performs better than these baseline methods, since we decompose the relation between ability and the language of the training corpus. In the previous model merging approaches, they mainly added the parameters of different models to obtain the final model, without considering the the relation between language and abilities. Due to the extraction mechanism of MAET, we mitigate the effect of languages and make the weight more related to ability, which can be transferred in multi-lingual scenarios.

## 5.3 DETAILED ANALYSIS

To comprehensively evaluate our proposed approach MAET and analyze its features, we conduct several experiments and detailed analysis in this part, including the ablation study, analysis of the transfer ratio of LLM parameters, and the generalization of MAET.

**Ablation Study.** To assess the effectiveness of each component of our proposed MAET, we conduct the ablation study and present the results in Figure 2. We implement MAET on multi-lingual math-

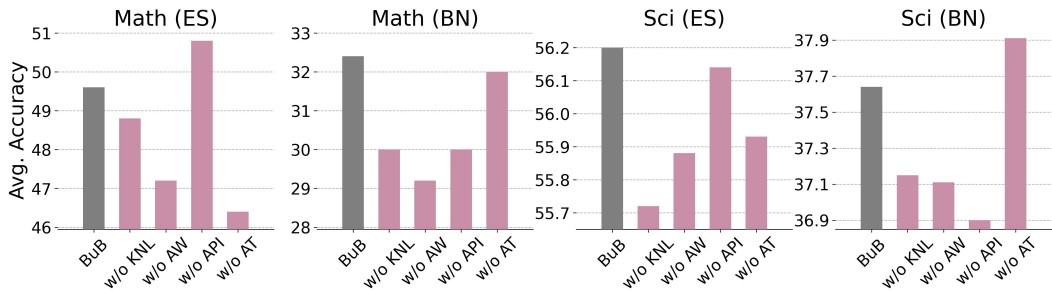

Figure 2: The results of ablation study. KNL, AW, API, and AT denote key neurons locating (Eq. 1), ability weights obtaining (Eq. 2), ability-related parameter tensors identifying (Eq. 4), and advanced abilities transferring (Eq. 5).

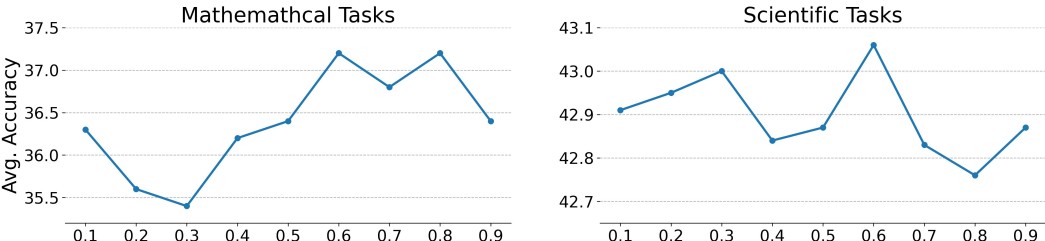

Figure 3: The performance of different proportions for the ability-related parameters identification.

ematical and scientific tasks without each module of MAET, *i.e.,* key neurons locating (*i.e.,* Eq. 1), ability weight obtaining (*i.e.,* Eq. 2), ability-related parameter tensor identifying (*i.e.,* Eq. 4), and advanced abilities transferring (Eq. 5). First, in most downstream scenarios, removing any module of MAET will affect the final performance, which has verified the effectiveness of the MAET process. Second, without ability weight obtaining, *i.e.,* directly utilizing the difference between LLM trained on the ability-related corpus and the backbone LLM as the ability weight, we can observe that the performance is seriously hurt in both scenarios, indicating this process can significantly extract the advanced abilities from the single-lingual corpus and decrease the influence of the language of the training corpus. Third, comparing the results of the models whether adopting the ability transferring process, experimental results show that LLM with the multi-lingual abilities enhanced cannot well solve multi-lingual mathematical and scientific tasks, and leveraging the ability weight provided by MAET can improve the LLM performance on advanced tasks.

**Ratio of Key Parameters During Transferring Stage.** Identifying and updating the ability-related sub-network of LLMs is the key point of our MAET. We conduct experiments to analyze the influence of the transferring ratio $k_2\%$ and show the results in Figure 3. Observing the results, the performance of LLM has decreased in a lower and higher ratio of the ability-related parameters identifying process. The main reason is that the lower proportion transfers incomplete knowledge to the model and makes LLM unable to possess the corresponding ability, affecting the performance on the downstream tasks. In contrast, the higher proportion cannot extract the ability weight precisely and will transfer too much language-related knowledge to the model, making the conflict with the LLM inner knowledge and hurting the multi-lingual scenarios performance.

**Ability-related Sub-networks of LLM.** To assess and probe the ability-related sub-networks of LLMs, we only transfer the specific tensors (*i.e.,* tensors in self-attention and MLP mechanism) from the ability weight to the final models through Eq. 5, to analyze the LLM inner abilities. The experimental results are presented in Table 3. From the experiment, we can observe that although the proportion of MLP layers (41.38%) is lower than the attention layers (45.26%), only transferring the MLP layers outperforms transferring the attention layers, indicating that the MLP layers are more related to the advanced abilities and stores the corresponding knowledge. In the MLP layers of LLM, the gate mechanism (*i.e.,* MLP Gate) will control the transmission of information and the

| LLM Tensors | Proportion of $\mathcal{T}$ | ES | ZH | BN | TE | Avg. |
|---|---|---|---|---|---|---|
| All Tensors | 100.00% | 49.60 | 41.60 | 32.40 | 25.20 | 37.20 |
| Attention All | 45.26% | 48.80 | 41.60 | 28.80 | 26.40 | 36.40 |
| Attention Q | 12.07% | 47.60 | 40.80 | 30.80 | 26.40 | 36.40 |
| Attention K | 10.34% | 47.20 | 42.40 | 29.60 | 24.40 | 35.90 |
| Attention V | 9.48% | 47.60 | 42.40 | 28.80 | 25.20 | 36.00 |
| Attention O | 13.36% | 48.00 | 40.40 | 30.80 | 27.20 | 36.60 |
| MLP All | 41.38% | 48.80 | 39.60 | 31.60 | 27.60 | 36.90 |
| MLP Up | 13.79% | 50.00 | 40.00 | 28.80 | 25.20 | 36.00 |
| MLP Gate | 13.79% | 46.00 | 41.20 | 30.00 | 24.00 | 35.30 |
| MLP Down | 13.79% | 49.60 | 41.60 | 30.40 | 26.00 | 36.90 |

Table 3: The effect of only merging the specific LLM tensors during the transferring process (*i.e.,* Eq.5) on multi-lingual mathematical tasks.

| Methods | MMLU | HumanEval | MBPP | OpenbookQA |
|---|---|---|---|---|
| LLaMA-3 8B | 60.85 | 35.98 | 46.60 | 65.00 |
| + CPT | 58.46 (-2.39) | 28.66 (-7.32) | 39.60 (-7.00) | 61.40 (-3.60) |
| + MAET (Ours) | 61.07 (+0.22) | 35.98 (+0.00) | 47.40 (+0.80) | 65.00 (+0.00) |

Table 4: The out-of-domain performance comparison of different training methods to train LLaMA-3 8B on OpenWebMath. During the ability-enhancing process, previous methods will hurt the OOD abilities of LLM, while our MAET can maintain the corresponding abilities.

down project mechanism (*i.e.,* MLP Down) will integrate the knowledge from previous layers, so that transferring the MLP layers can achieve better performance on the downstream tasks.

**Out-of-Domain performance of MAET.** We conduct experiments about adapting mathematical ability on the general LLM through MAET, and assess the performance on out-of-domain (OOD) tasks (*i.e.,* MMLU (Hendrycks et al., 2021), HumanEval (Chen et al., 2021a), MBPP (Austin et al., 2021), and OpenbookQA (Mihaylov et al., 2018)), which can reflect and assess different abilities of LLMs. Results are presented in Table 4. We can observe that the performance of LLM on all evaluation tasks has decreased through the CPT training process, and the maximum decrease has been achieved 7.32% on the HumanEval task. One of the possible reasons is that LLaMA-3 has been trained on OpenWebMath during pre-training and the CPT process makes it overfit and forget the knowledge of other domains, hurting the performance on OOD tasks. In contrast, our proposed MAET achieves comparable and even better performance with backbone LLM in all downstream scenarios. Since we identify and update the key neurons related to the specific ability, the ability of LLM can be precisely enhanced, and this strategy also helps the OOD tasks needed for mathematical ability, *e.g.,* mathematical sub-tasks in MMLU and code synthesis task MBPP.

## 6 CONCLUSION

In this paper, we presented MAET, which extracted the advanced ability-related weights from the LLM and supported simple addition and subtraction operations to transfer the ability across different languages. Concretely, MAET included two main stages, *i.e.,* extraction and transfer. For the extraction stage, we located the key neurons and extracted the ability-related weights. Then, in the transfer stage, we identified the key parameter tensors and leveraged them to transfer the advanced ability into other LLMs. In this process, the multi-lingual ability-related training corpus is not required, and the experimental results have shown that our approach outperformed competitive baselines.

As future work, we will consider better methods to identify the ability-related sub-network to decompose the abilities of LLMs and utilize an automated approach to determine the hyper-parameter. Besides, we will implement MAET on larger-scale models, and scenarios with more languages and requiring more abilities to evaluate its effectiveness.

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

| Concepts | Meaning |
|---|---|
| Key Neurons | Neuron refers to one of the trainable values of the tensors in LLMs. As previous work pointed out (Xu et al., 2024a), different neurons might control the different abilities of LLMs. Following this finding, in our work, we define the neurons that control the specific ability as the "Key Neurons". Key neurons can be regarded as a set without duplication, and a neuron belonging to the set means that this neuron can control the specific ability (Chen et al., 2024b). During the following training process, only the neurons belonging to the key neurons will be trained and optimized. |
| Ability-related Weights | Ability-related weights refer to the value of the whole neuron in LLM, which can represent the corresponding ability of LLM (Yu et al., 2024a; Ilharco et al., 2023). In MAET, we obtain the ability-related weights through equation 2. The ability-related weights contain the value of all neurons. Since only the key neurons will be trained during the training process, the value of the neurons not belonging to key neurons is zero in the ability-related weights. |
| Ability-related Tensors | Ability-related tensors can be regarded as a set of LLM tensors, which is related to the corresponding ability. Previous work has studied how the LLM layers influence the ability (Cheng et al., 2024). Different from key neurons, ability-related tensors focus on higher-level information, integrating the sparse neurons into a coarser-grained element (Xiao et al., 2024). A tensor belonging to the ability-related tensors denotes that this tensor is highly related to the corresponding ability and can control this ability. |
| Language-specific Weights | Similar to the ability-related weights, language-specific weights also refer to the value of the whole neurons in LLMs (Zhang et al., 2024b). However, language-specific weights represent the language abilities of LLM that include multiple abilities (i.e., one language can be regarded as one ability) (Tang et al., 2024), and the method of obtaining them is also different from ability-specific weights. In MAET, we first calculate the ability-related weights of each language and then Integrating these weights together to obtain the language-specific. |

Table 5: The key concepts of our approach.

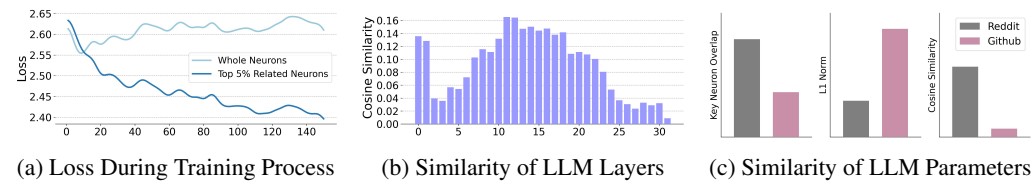

(a) Loss During Training Process     (b) Similarity of LLM Layers     (c) Similarity of LLM Parameters

Figure 4: The results of empirical experiments. We present the loss of different training methods during the training process, the cosine similarity of LLM layers after being trained on Zhihu and Reddit, and the similarity of LLMs being trained on different training corpus.

## A  EMPIRICAL STUDY

A surge of work (Zhang et al., 2024b; Xiao et al., 2024; Tang et al., 2024) has pointed out that LLMs sparsely activate the specific sub-modules to perform corresponding tasks. Based on these findings, we conduct empirical experiments to explore whether the specific sub-module, which is related to advanced abilities, can be extracted and transferred. We utilize the forum corpus (*i.e.,* Zhihu for Chinese forum corpus and Reddit for English forum corpus) to continually pre-train LLMs, and then assess the training performance (*i.e.,* the value of loss function) and similarity of LLM neurons.

The forum corpus can be considered as containing the question-answering (QA) ability, which is necessary and important for LLMs. The results from Figure 4a have shown that only training the top 5% relevant neurons of LLMs can achieve the lower training loss and fit into the training set more quickly, indicating that LLMs contain the sub-module corresponding to the QA ability. Moreover, from Figure 4b and Figure 4c, we can observe that the LLM trained on Zhihu has shown higher

similarity with the LLM trained on Reddit than the LLM trained on Github (*i.e.,* lower L1 Norm and higher cosine similarity), and the cosine similarity of different layers in LLM are largely different.

According to the above results, we have found that the different sub-networks of LLMs control the different abilities, and precisely selecting the correct sub-module of LLMs will help the extraction of advanced abilities from the single-lingual corpus and the transfer of these abilities to multi-lingual scenarios. Concretely, although Zhihu and Reddit are in different languages, they will influence the similar sub-modules of LLM and make these sub-networks show high similarity with each other. These sub-networks can be referred to the ability-related sub-networks, which are slightly influenced by languages.

| Stage | Hyper-Parameter | Mathematical Tasks | Scientific Tasks |
|---|---|---|---|
| | Learning Rate | $5 \times 10^{-5}$ | $5 \times 10^{-5}$ |
| | Batch Size | 1M Tokens | 1M Tokens |
| | Training Steps | 2B Tokens | 2B Tokens |
| Extracting | $\alpha$ in Extraction | 0.8 | 0.8 |
| | $\beta$ in Extraction | 0.2 | 0.2 |
| | Ratio of Key Neurons | 5% | 5% |
| | Learning Rate | $5 \times 10^{-5}$ | $5 \times 10^{-5}$ |
| | Batch Size | 1M Tokens | 1M Tokens |
| | Training Steps | 2B Tokens | 2B Tokens |
| | $\gamma$ in Transferring | 0.2 | 0.2 |
| | $\eta$ in Transferring | 1.0 | 1.0 |
| Transferring | Ratio of Key Neurons | 80% | 60% |
| | $\mu$ for Spanish | 1.5 | 1.5 |
| | $\mu$ for Chinese | 2.0 | 2.0 |
| | $\mu$ for Bengali | 1.2 | 1.2 |
| | $\mu$ for Telugu | 1.2 | 1.2 |

Table 6: The details of hyper-parameters in the training and evaluation process.

| Language | Training Dataset (Tokens) | | Evaluation Dataset (Instances) | |
|---|---|---|---|---|
| | General Corpus | Ability-related Corpus | Mathematical Tasks | Scientific Tasks |
| English | 1.81B | 1.30B (Math) / 1.82B (Sci) | 250 | 1,245 |
| Spanish | 1.81B | - | 250 | 1,232 |
| Chinese | 1.80B | - | 250 | 1,229 |
| Bengali | 1.81B | - | 250 | 1,137 |
| Telugu | 1.81B | - | 250 | 1,036 |

Table 7: The statistical information of the training and evaluation datasets.

| Methods | Qwen2.5 0.5B | | | Gemma2 2B | | |
|---|---|---|---|---|---|---|
| | ES | TE | Avg. | ES | TE | Avg. |
| Backbone LLM | 36.64 | 25.69 | 31.17 | 43.41 | 30.01 | 36.71 |
| + F-CPT$_{L\&A}$ | 32.90 | 22.43 | 27.67 | 38.48 | **30.39** | 34.62 |
| + F-CPT$_A$ | 32.62 | 25.26 | 28.94 | 37.83 | 25.39 | 31.61 |
| + MAET w/o API | 36.72 | 28.91 | 32.82 | 43.23 | 29.59 | 36.41 |
| + MAET (Ours) | **36.91** | **29.62** | **33.27** | **43.62** | 30.37 | **37.00** |

Table 8: The performance comparison of different LLMs on multilingual scientific tasks.

# B DETAILS OF HYPER-PARAMETERS

We release all of the hyper-parameters during our experiment to better reproduce our proposed approach. Table 6 shows the details of hyper-parameters of different stages.

## C  DETAILS OF DATASET

We present the statistical information of the datasets in Table 7. We mainly consider English, Spanish, Chinese, Bengali, and Telugu in our experiment, and utilized English as the in-domain language while others as the out-of-domain languages. For the evaluation datasets, we select MGSM and multi-lingual MMLU as the evaluation benchmarks, which contain the parallel data in different languages and are useful for multi-lingual complex tasks evaluation.

## D  PROMPT FOR TRANSLATION

```
You should translate the following text from English to {TARGET
LANGUAGE} and should not modify the latex code or website code.
You should not add any details that are not mentioned in the
original text.

## English
{ENGLISH TEXT}

## {TARGET LANGUAGE}
```

## E  PERFORMANCE OF SMALL SCALE LLMs

We conduct the different LLMs with different sizes (*i.e.*,, Qwen2.5-0.5B and Gemma2-2B) in our experiment to valid the practicality of our approach. We assess MAET and baselines on multi-lingual scientific reasoning tasks and present the evaluation results in Table 8. Comparing the performance of MAET and the baseline methods, we can observe that MAET can also enhance the performance of small scale models and outperform competitive baselines. Therefore, the evaluation results have shown the effectiveness of MAET and verified that MAET is a general LLM enhancement technology.

## F  LIMITATIONS

In this section, we discuss the limitations of our work. First, we only implement our approach MAET on 8B LLMs (*i.e.,* LLaMA-3 8B), and do not adopt the LLMs with larger scales (*e.g.,* 13B or 70B LLMs) in the experiment, due to the limitation of computational resources. We will test the effectiveness of our approach on these LLMs in the future. Second, we only evaluate our approach on two downstream tasks (*i.e.,* mathematical and scientific reasoning tasks) in multi-lingual scenarios. Although they are challenging and widely-used testbeds, it is still meaningful to verify our methods on other tasks. Whereas, as we test the performance on diverse high-resource and low-resource languages, it can also provide comprehensive performance estimation for our approach in multi-lingual scenarios. Third, our approach is a general method for ability transferring across different domains, but in this work, we only consider the multi-lingual scenarios and obtain a multi-lingual LLM with the specific ability being enhanced. Forth, our approach also relies on continual pre-training the LLM. Although the training corpus is not very large, it also increases the cost. Fortunately, after we obtain the pre-trained weights, our following steps only need simple addition and subtraction operations for ability transferring, which is flexible for online deployment and application. In future work, we will focus on reducing the data requirement for the pre-training corpus and also improving the weights extracting efficiency. Finally, we do not consider the potential risk and ethics issues that might hurt the alignment of LLMs when using our approach. Actually, our approach is also applicable to transfer the alignment ability across languages. We will investigate to it in the future.

