# OpenReview forum: "Extracting and Transferring Abilities For Building Multi-lingual Ability-enhanced Large Language Models"
_ICLR.cc/2025/Conference — ICLR 2025 Conference Withdrawn Submission_

### Official Review · Reviewer_tAan · 2024-11-01

**Soundness:** 3
**Presentation:** 2
**Contribution:** 2
**Rating:** 6
**Confidence:** 4

**Summary:**

This paper seeks to combine multilingual abilities and other "abilities" together when ability-specific data is not available in a particular low-resource target langauge. In order to do so, they propose a solution that involves first finding the parameters that are most important for the "ability" by finding the most important continually pretraining separate variants of a base LLM, one on generic multilingual data and the other on ability-specific data, typically in English. Then, they extract what parameters are holding the learned information by evaluating which parameters changed most for each. Using these parameters, they then merge the two variants together to create a model that has boosted capabilities in the target task in the target language. They evaluate this method across numerous LLMs, two tasks (math & science), four languages, and evaluate numerous baselines, all of which show the effectiveness of MAET.

**Strengths:**

1. This paper extends previous work on task vectors and outline a novel methodology to extract parameter updates most important to language capabilities and to a specific domain/task. This meticulously designed methodology then allows for combining model updates in a matter that alleviates parameter interference.
2.  The experiments are impressively thorough: they cover multiple models, two tasks, many possible CPT setups, and a small, but diverse set of languages. This extensive experimentation clearly took significant resources and effort. The results show MAET is effective and comparable to different possible full CPT setups.
3. The results show that MAET is significantly more effective than the well-established model merging method from Ilharco et al., 2023, which does arithmetic over task vectors.
4. Interesting secondary findings, which they detail in Section 5.3.
a) One is the parameter location that they find to be most important (Line 474-482). This aligns with previous literature, but they come to this conclusion in a very different manner
b) The second is the fact that MAET is much more successful in not "forgetting" other domains/tasks in comparison to full CPT. This would need further experimentation to control for overfitting in both settings, but this is a very promising avenue for why MAET would be impactful.

**Weaknesses:**

1. Shortcomings of neuron locating strategy:

a) I don't think that the value change of neurons is an appropriate approximation for accumulated gradients. The value change maps the net trajectory taken after many gradient updates, versus the accumulated gradient is the mean gradient across different data points. At minimum, you could remove mention of accumulated gradients and state that value change is the desired method for estimating neuron importance.

b) (Line 205) I don't understand why you would train \nu_L_0 and create LLM_{A_i,L_0}. Would it not make sense to simply subtract the language difference from the ability difference and not from the ability & language difference.

c) Is R(A_i) therefore a task vector as in Ilharco et al., 2023 ? This should be mentioned as it is currently very unclear what R(A_i) represents.

d) [Most important] Calculating R(A_i) required ability-specific and generic data in English (L_0). In that case, it is very unclear what R(L) is for all other languages and how it was calculated. I thought domain/ability-specific data for these target languages was unavailable ? Authors need to define how exactly R(L) is calculated for languages other than English and if ability-specific data in the target language is used.

e) Generally, this section lacks justification and references to justify the methodology, many parts of which feel arbitrary.

f) In Line 161, the authors imply they use a small subset of C_{L_0,A_i} and yet Algorithm 1 implies all available data samples in C_{L_0,A_i} are used.

3. This paper positions this method as one to use when data in the target languages is highly limited, and yet they use 1M tokens per language. This is a significant amount of in-language data, undermining the practicality of this method.
4. Results are not very impressive. Full continual pre-training on all the data available is a much simpler and less computationally expensive method and it performs very close on average. I would argue this average performance is within a hyperparameter-tuning error range — by that I mean is there is likely hyperparameter setups where full continual pre-training data is slightly better across languages and therefore the average is higher than MAET. The same applies to full CPT on just the multilingual data + domain adaptation. I would need more evidence of full effort to maximize performance in comparative scenarios (i.e. CPT scenarios).

a) Why is CPT only on the ability data not evaluated ? My intuition is that that would be more effective than CPT on the multilingual generic data only

5. The authors have interesting findings in Section 5.3, but do not highlight them and present them as key contributions of the paper. I believe that a possible reformulation of the presentaiton of this work which puts these findings more in focus would lead to a more captivating paper.

**Questions:**

- I don't recommend the term "ability"/"ability-related" that the authors selected. I would use the term "domain"/"domain-specific" as it is more common and more concise to what you are attempting to do: improve a model's capability in a target domain in a target language.

- Line 43: "dominated the linguistic expressions of the open web data" does not make sense. I would word this differently.
- Line 44, 131: In Bengali and Telugu, there are millions (maybe even billions of samples) available on the open web. I would specify this statement that either the availability of (a) *labeled* data or (b) domain-specific data is highly limited. I think the framing of these languages as low-resource is a stretch given they are some of the 10-20 most spoken languages in the world. I would encourage the authors to emphasize the
- Line 45: I would include older citations for cross-lingual transfer, a phenomena that has been studied for 5+ years.
- Line 56: Please explain "multilingual ability-enhanced LLM *like building blocks*".
- Line 70: improvement *compared to the base* LLM
- Line 74,336,366: "monolingual" instad of "single-lingual"
- Line 116-123: there are numerous typos
- Line 120: "Despite the fact that"
- Line 160: How small of an amount ?
- Line 197: You need to recite the TIES-merging paper here
- Line 246: How are you calculating the cosine similarity of a multi-dimensional tensor ? Are you flattening it before calculating the dot product ?
- Line 250: Is sim() the dot product or cosine similarity ?
- Line 336: Use of "L" to mean both LoRA and multilingual general corpus is very confusing
- Line 370: You need to cite the LoRA paper
- Table 3: So if i understand correctly, only merging these specific tensor types individually get you almost the same score ? This seems like a very significant finding, no ?
- Line 470-472: This claim is not explained or justified with references.
- Line 474-482: This also seems like a more significant finding than authors give it credit for.
- Line 539: You need to provide justification for the claim that MAET is applicable to safety alignment.
- Stylistic suggestion: I believe you are overusing "e.g.", for the purpose of reading flow, I would suggest using alternative terms "like", "such as", and sometimes i.e. instead of e.g.

- an high-impact body of work that is highly related yet not discussed is Composable Sparse Fine-Tuning for Cross-Lingual Transfer by Ansell et al.

---

> ### Author Response · Authors · 2024-11-22
> **Response to the Weakness - Part 1**
>
> > Shortcomings of neuron locating strategy
>
> **For the most important issue**, in the ability-related parameter tensor selection step of the transferring stage, we utilize the multi-lingual general corpus to train the LLM the corresponding neurons and then calculate the ability-related weights of languages R(L_i), without training model on the multi-lingual ability-related corpus. The language-specific weights are calculated from other languages R(L_i), and the ability-specific data in the target language is not utilized in the whole process of MAET. Actually, we can easily collect the multi-lingual general corpus from the website, but it is hard to collect the ability-related corpus in multi-lingual scenarios. Thus, in this work, we follow the real-world limitations to conduct our approach and experiment.
>
> **For the neuron locating strategy**, a surge of work[1][2][3] has pointed out that different neurons of LLM might control different abilities, and only activating or freezing a series of neurons can maintain or hurt the performance of LLMs on the corresponding tasks. Based on these findings, we wonder how to select the neurons that high-related to the specific ability. According to the influence formulation[4], the gradient can reflect the importance of training data. Inspired by previous work [5][6][7], the value of the gradient of each LLM neuron from a single training instance can reflect the influence of this training instance on the corresponding neuron, and the neuron with a small gradient can be sheared in the training process. Therefore, we adopt the accumulated value of the gradient to identify the key neurons, and the experimental results have shown the effectiveness of our locating strategy.
>
> [1] The lottery ticket hypothesis: Finding sparse, trainable neural networks.
>
> [2] Sharing matters: Analysing neurons across languages and tasks in llms.
>
> [3] Unveiling Linguistic Regions in Large Language Models.
>
> [4] Estimating training data influence by tracing gradient descent.
>
> [5] LESS: selecting influential data for targeted instruction tuning.
>
> [6] Not Everything is All You Need: Toward Low-Redundant Optimization for Large Language Model Alignment.
>
> [7] Unlocking Continual Learning Abilities in Language Models
>
> **For our design of the approach**, our basic idea is to learn and extract the ''ability-related weight'', which can achieve building a powerful model more effectively and efficiently. Based on this idea and inspired by previous work, we decompose the whole process into the extracting stage and the transferring stage. In the extracting stage, we first locate the key neurons that high-related to the downstream ability and then train LLM on these selected neurons. Since ability A_i is in language L_0, we train LLM on the merge of A_i-related key neurons and L_0-related key neurons, to further reduce the influence of language L_0 and make it more easier to extract the ability-related. And in the transferring we observe that several tensors in ability-related weights have shown high-similarity with language-specific weights, indicating that the knowledge of language still stored in the ability-related weights. Thus, we adopt Eq.4 to further remove the effect of the language and purify the ability-related weights that are more suitable for the multi-lingual scenarios. Experimental results in Table 2 and Figure 2 have shown the effectiveness of our design.
>
> For the typos, we thank the reviewer have carefully reading our paper, and will update it in the next revision.
>
> > This paper positions this method as one to use when data in the target languages is highly limited, and yet they use 1M tokens per language. This is a significant amount of in-language data, undermining the practicality of this method.
>
> Actually, in real-world practice, the general corpus of different languages can be easily collected from the internet, like Wikipedia or corresponding forums. However, the ability-related corpus (e.g., mathematics-related corpus or science-related corpus) is highly limited, since lacking collecting and organizing. Given this fact, in our work, we utilize the **multi-lingual general corpus** and only **single-lingual ability-related corpus** to construct our training dataset. Because of the highly limited ability-related corpus, we do not adopt the ability-related in other languages, like Bengali or Telugu. Our approach and experiment setting strictly follow the real work conditions, and the experiment results have shown the effectiveness and practicality of our method.

---

> ### Author Response · Authors · 2024-11-22
> **Response to the Weakness & Question - Part 2**
>
> > Results are not very impressive. Full continual pre-training on all the data available is a much simpler and less computationally expensive method and it performs very close on average. I would argue this average performance is within a hyperparameter-tuning error range — by that I mean is there is likely hyperparameter setups where full continual pre-training data is slightly better across languages and therefore the average is higher than MAET. The same applies to full CPT on just the multilingual data + domain adaptation. I would need more evidence of full effort to maximize performance in comparative scenarios (i.e. CPT scenarios).
>
> In this work, we focus on how to combine a set of abilities by using the existing elements to efficiently obtain the multi-lingual ability-enhanced LLM,  under the limitation of multi-lingual ability-related training corpus. Performance on downstream tasks and computation expense are the issues that we are most concerned with. Compared with full CPT scenarios, our methods can efficiently adapt to new a domain with less computation cost and achieve better performance.
>
> Concretely, to adopt LLM to a new domain, full CPT should mix the multi-lingual general corpus and the new single-lingual ability-related corpus, and train LLM on this large-scale dataset. In contrast, since the language-specific weights can be reused in different downstream tasks, MAET only trains the model on the new single-lingual ability-related corpus to obtain the ability-related weights, and then utilizes the ability-related weights and the pre-processed language-specific weights to build the final LLM. The computation consumption of MAET is significantly lower than full CPT.
>
> Besides, we have searched for the best hyper-parameters for all baselines. In this situation, MAET still achieves the best performance in an average score overall baseline methods.
>
> To conduct a more comprehensive and persuasive experiment, we employ our approach on other foundation LLMs and assess its performance on multi-lingual scientific reasoning tasks. As shown in the following table, comparing the performance of MAET and the full CPT, we can observe that the CPT methods cannot transfer the ability from English to Spanish and Telugu, while MAET can enhance the performance of LLMs on multi-lingual scenarios, especially in low-resource language Telugu. The following results have shown the effectiveness of our MAET.
>
> | Methods | ES | TE | Avg. |
> |:--|:--:|:--:|:--:|
> | Qwen2.5-0.5B | 36.64 | 25.69 | 31.17 |
> | + CPT$_{L \\& A}$ | 32.90 | 22.43 | 27.67 |
> | + CPT$_{A}$ | 32.62 | 25.26 | 28.94 |
> | + MAET w/o API | 36.72 | 28.91 | 32.82 |
> | + MAET | **36.91** | **29.62** | **33.27** |
>
> | Methods | ES | TE | Avg. |
> |:--|:--:|:--:|:--:|
> | Gemma2-2B | 43.41 | 30.01 | 36.71 |
> | + F-CPT$_{L \\& A}$ | 38.48 | **30.39** | 34.62 |
> | + F-CPT$_{A}$ | 37.83 | 25.39 | 31.61 |
> | + MAET w/o API | 43.23 | 29.59 | 36.41 |
> | + MAET | **43.62** | 30.37 | **37.00** |
>
> > The authors have interesting findings in Section 5.3, but do not highlight them and present them as key contributions of the paper. I believe that a possible reformulation of the presentaiton of this work which puts these findings more in focus would lead to a more captivating paper.
>
> In this work. we focus on conducting a more low-cost and effective ability transfer. Based on this major gold, we wonder whether the ''ability-related weights'' can be learned and extracted, and design the MAET approach, which first extracts the ''ability-related weights'' and then leverages these weights to construct a multi-lingual ability-enhanced LLM like building blocks. The approach MAET can effectively and efficiently transfer the abilities from single-lingual scenarios to multi-lingual scenarios, without being trained on the multi-lingual ability-related corpus, which strictly follows the real work conditions. Thus, MAET can be regarded as one of the key contributions of out work and is the most important aspect we want to share.
>
> To further analyze the future of our approach MAET, we present the details analysis in section 5.3, including the ratio of key parameters during the transferring stage, ability-related sub-network of LLM, and out-of-domain performance of MAET. The discussion and experiments have explored the internal mechanisms of the model's control over its abilities and preliminarily revealed the relation between abilities and the sub-network of LLM. These findings are also important contributions of our paper.
>
> Thanks for the reviewer's suggestions, we will reorganize the content in our paper and highlight the interesting findings of our paper, to make it more strong and captivating.
>
> **Response to the question**
>
> Thanks for carefully reading our paper and providing valuable suggections. These questions are about the expression, citation, style, and the logic of the paper. We will modify these questions in the next revision of our paper.

---

> ### Comment · Reviewer_tAan · 2024-11-26
> **Response to Author Rebuttal**
>
> Thanks for the many clarifications and comments. I will upgrade overall assessment 5->6, but downgrade confidence from 5->4. The upgrade to 6 was very close, and I think the paper still has a few shortcomings that need to be addressed. Most notably:
>
> - I still believe the presentation of the work, notably Section 4, is lacking in clarity.
> - The results don't represent a standout improvement given the complication of the method, limiting the potential adoption.

---

> > ### Author Response · Authors · 2024-11-27
> > **Appreciate for raising score And Response for the reviewer's concerns**
> >
> > Dear Reviewer  tAan,
> >
> > Thank you so much for raising the score. We would greatly appreciate the opportunity to engage in discussions with you. And the following is our response to your concerns, and hope these responses can address your concerns about our paper.
> >
> > > I still believe the presentation of the work, notably Section 4, is lacking in clarity.
> >
> > To further demonstrate our method and help readers better understand our approach, we have uploaded a new revision of our paper, and presented a discussion in Section 4.3, including the table to clarify the key concepts in our paper and the pseudo code for our approach.
> >
> > In our paper, there are four key concepts, i.e., Key Neurons, Ability-related Tensors, Ability-related Weights, and Language-specific weights. ''Key Neurons'' and ''Ability-related Tensors'' denote LLM inner neurons and tensors, which are related to the corresponding abilities. And ''Ability-related Weights'' and ''Language-specific weights'' denote the value of the whole neurons of LLMs, which can control the abilities or languages.
> >
> > The procedure of MAET consists of two main stages, i.e.,  ability-related weights extraction and multi-lingual ability transferring. For the extraction stage, we first utilize the accumulated gradient to estimate the importance of each neuron by Equation 1. Then, we leverage the model trained on the general corpus to remove the influence of language and obtain the ability-related weight through Equation 2. In the transfer stage, we utilize Equation 3 and Equation 4 to obtain the multi-lingual weight and identify the ability-related parameter tensors in LLM. After that, we leverage Equation 5 to fulfill the multi-lingual ability transfer, to build the multi-lingual ability-enhanced LLM.
> >
> > > The results don't represent a standout improvement given the complication of the method, limiting the potential adoption.
> >
> > In comparison to the performance of the backbone model LLaMA3-8B, employing MAET significantly enhances its capabilities in multilingual complex reasoning scenarios, achieving a relative improvement of 9.1%. To facilitate the reproduction and adoption of our approach, we have submitted the code as supplementary material and will make it publicly available.
> >
> > For the baseline methods, like $F-CPT_{L\\&A}$, they need to utilize the language-specific corpus and ability-related corpus to train a new model, which will consume a large computational resource in both the training process and deploying process. However, In MAET, we only need a small amount of computational resources to adapt the language model to the new domain and new language and outperform the CPT methods. This is an exciting contribution to adapting LLMs to new scenarios, particularly in low-resource settings with limited training data and computational resources.

---

### Official Review · Reviewer_11CH · 2024-11-03

**Soundness:** 3
**Presentation:** 3
**Contribution:** 3
**Rating:** 8
**Confidence:** 3

**Summary:**

The paper presents a novel solution towards enhancing a specific ability for low-resource languages. Their method MAET requires a multi-lingual general corpus and an ability-specific corpus from a single language (e.g., English). After extracting ability-related weights, the related parameter tensors can be readily patched to improve multi-lingual performance. Results and ablations have shown the effectiveness of MAET. Analysis provides insights onto the possible location of ability-related parameters.

**Strengths:**

- **Novelty**: key units are identified to implement sparse update in model training and merging.
    - In the extraction stage, key neurons and ability-specific weights are identified.
    - In the transfer stage, ability-related parameters are further filtered by their similarity with language-related parameters.
- **Concrete experiments**: ablations establish the necessity of each component in MAET.
- **Insightful analysis**: the detailed analysis provides insights.

**Weaknesses:**

- A useful and strong baseline would be to include a multi-lingual ability-related corpus for training.
    - This can be obtained by translating $C_{{L_0}, A_i}$ to $C_{{L_j}, A_i}$ for other languages $j$.

**Questions:**

- Can you explain how domain adaptation is done?

---

> ### Author Response · Authors · 2024-11-22
> **Response to the Weakness**
>
> We sincerely thank the reviewer for the insightful suggestions.
>
> > A useful and strong baseline would be to include a multi-lingual ability-related corpus for training. This can be obtained by translating C_{L0,Ai} to C_{Lj,Ai} for other languages j.
>
> Translating single-lingual corpus to multi-lingual corpus can obtain the ability-related corpus in multi-lingual scenarios. However, during the translation process, a strong translator is highly required and previous work has widely utilized GPT-4o or Claude 3 as the translator to translate the training corpus. This kind of approach will bring two critical problems:
>
> + The first is the large consumption of resources including GPU, money, and time. The utilization of the open-source model will require GPU resources and the utilization of the closed-source model will expense much money to access API. And the translation process will also cost much time.
> + The second one is the low quality of the translated corpus. Although the powerful LLMs have been trained on multi-lingual corpus and can perform well in multi-lingual scenarios, the translation might contain the noise and mistakes. These unexpected components in the translated corpus will affect the performance of LLM trained on them.
>
> Despite the disadvantage of the translation-based methods, following the suggestion from reviewers, we adopt GPT-4o to translate the ability-related corpus from English to Spanish, Chinese, Bengali, and Telugu, and utilize the translated corpus to continually pre-train the LLM. In our experiment, CPT_{T}, CPT_{S & T}, and CPT_{S} denote continually pre-train the LLM on translated multi-lingual corpus, original single-lingual corpus and translated multi-lingual corpus, and original single-lingual corpus, respectively. The evaluation results are presented in the following table.
>
> | Methods | ES | ZH | BN | TE | Avg. |
> |:--:|:--:|:--:|:--:|:--:|:--:|
> | Backbone | 55.06 | 47.24 | 36.63 | 29.26 | 42.05 |
> | + CPT_{T} | 50.35 | 45.36 | 34.54 | **34.46** | 41.18 |
> | + CPT_{S \& T} | 53.73 | 46.30 | 35.06 | 31.73 | 41.71 |
> | + CPT_{S} | 51.90 | 45.71 | 33.35 | 29.41 | 40.09 |
> | + MAET | **56.20** | **48.00** | **37.64** | 30.38 | **43.06** |
>
>
> According to the results in the above table, we can observe that continual pre-training LLM on the translated multi-lingual corpus cannot effectively enhance the performance of LLM in all language scenarios and continual pre-training LLM on the original single-lingual corpus will make LLM overfit to the English scenarios and loss the abilities in multi-lingual scenarios. In contrast, our MAET can alleviate the influence of different languages and transfer the ability from English to multi-lingual scenarios.

---

> ### Author Response · Authors · 2024-11-22
> **Response to the Question**
>
> > Can you explain how domain adaptation is done?
>
> In our approach MAET, we utilize the single-lingual ability-related corpus and multi-lingual general corpus to extract the corresponding ability and transfer this ability from single-lingual scenarios to multi-lingual scenarios. Concretely, MAET includes the extracting and transferring stage, where extract the ability-related weights and transfer of the ability-related to build the final LLM, respectively. There are several key concepts that should be noticed, i.e., Key Neurons, ''Ability-related Weights'', ''Language-specific weights'', and ''Ability-related Tensors''. ''Key Neurons'' and ''Ability-related Tensors'' denote LLM inner neurons and tensors, which are related to the corresponding abilities. And ''Ability-related Weights'' and ''Language-specific weights'' denote the value of the whole neurons of LLMs, which can control the abilities or languages.
>
> First, in the extracting stage, we first locate the key neurons and utilize the accumulated value of the gradient, and then we continually pre-train LLM on the training corpus only on these selected key neurons and utilize Eq.2 to obtain the ability-related weights. The ability-related weights contain information on the relation between each neuron in LLM and the corresponding ability.
>
> Next, in the transferring stage, to further remove the influence of language, we identify the ability-related tensor by calculating the cosine similarity between ability-related weights and language-specific weights through Eq.4. And we transfer these language-agnostic weights (i.e., the selected tensor) from ability-related weights to language-specific weights, building the multi-lingual ability-enhanced LLM.
>
> After extracting and transferring stage, the LLM can adapt to the target domain in multi-lingual scenarios.

---

> ### Author Response · Authors · 2024-11-28
> **Kindly Reminder**
>
> Dear Reviewer 11CH,
>
> Thanks for your careful reading of our paper and the positive scores. We carefully explained your concerns and tried our best to elaborate the unclear points. We would like to know whether you find our response satisfactory, or if there are more questions that we could clarify. We are more than happy to hear your comments and address any of your further concerns during the remaining time.
>
> Best,
>
> Authors

---

> ### Comment · Reviewer_11CH · 2024-11-29
>
> Thanks so much for the effort and clarification. I decided to raise the score from 6 to 8.

---

> > ### Author Response · Authors · 2024-11-29
> > **Appreciate for raising score**
> >
> > Dear Reviewer 11CH,
> >
> > Thank you so much for raising the score. We would greatly appreciate the opportunity to engage in discussions with you. if you have any other questions, we hope our response can assist you in addressing your concerns.
> >
> > Best,
> >
> > Authors

---

### Official Review · Reviewer_4Vk6 · 2024-11-05

**Soundness:** 3
**Presentation:** 3
**Contribution:** 3
**Rating:** 5
**Confidence:** 4

**Summary:**

This paper presents a novel method to enhance the multilingual capabilities of large models, providing the model with new abilities through both the extraction and transfer stages. In the extraction stage, MAET sequentially identifies key neurons and learns ability-specific weights. In the subsequent transfer stage, MAET selects ability-related parameter tensors, which are then used to construct a multilingual ability-enhanced LLM. Experiments have shown that MAET is better than some existing continuous pre-training methods in multilingual mathematics and science tasks.

**Strengths:**

-	A novel multilingual enhancement scheme that offers valuable inspiration for future research.
-	The technical solution is sound, and the details, such as the method for cumulative gradient, are well thought out.

**Weaknesses:**

-	Although the technical contribution of this paper is novel, the experiments are relatively limited and do not fully validate the proposed contributions. For instance, the paper only tests the effect of using LLaMA-3 as the backbone, but as a general LLM enhancement technology, this scope is insufficient. I understand that training large models requires significant resources, but there are many existing smaller LLMs worth experimenting with, such as Qwen2.5-0.5/1.5B and Gemma2-2B.
-	The choice of some hyperparameters is not clear, such as \mu in equation (3).
-	This paper involves many abbreviations. It is recommended to bold the first appearance of the baseline and dataset names in Section 5.1 for clarity.
-	The related work on model merging is missing key citations [1][2], which demonstrate significantly better performance on multilingual mathematical tasks (MGSM dataset). The authors should reference these studies and engage with their findings in the discussion.

[1] LangBridge: Multilingual Reasoning Without Multilingual Supervision

[2] MindMerger: Efficient Boosting LLM Reasoning in non-English Languages

**Questions:**

-	Is the “Ability-related Parameter Tensor Selection” in Section 4.2 more closely related to the extraction stage in Section 4.1 rather than the transfer stage?
-	The concepts of “tensor selection” and “parameter weights” are easily confused. Can add examples or annotate the dimensions to distinguish them?

---

> ### Author Response · Authors · 2024-11-22
> **Response to the Weakness**
>
> We sincerely thank the reviewer for the insightful suggestions.
>
> > Although the technical contribution of this paper is novel, the experiments are relatively limited and do not fully validate the proposed contributions. For instance, the paper only tests the effect of using LLaMA-3 as the backbone, but as a general LLM enhancement technology, this scope is insufficient. I understand that training large models requires significant resources, but there are many existing smaller LLMs worth experimenting with, such as Qwen2.5-0.5/1.5B and Gemma2-2B.
>
> In our evaluation process, we also consider strong baselines (e.g., continually pre-training, model merging) and adapt our method on the various downstream tasks (i.e., mathematical reasoning tasks, and scientific reasoning tasks) in multi-lingual scenarios (i.e., Spanish, Chinese, Bengali, and Telugu). According to the experimental results, our approach MAET has achieved the best performance on the average score of all the tasks. These results have shown the effectiveness of our method.
>
> To conduct a more comprehensive experiment and strengthen our paper, we follow the suggestions from the reviewer and test our approach on small models (i.e., Qwen2.5-0.5B and Gemma2-2B) into our experiment. We assess MAET and baselines on multi-lingual scientific reasoning tasks and present the evaluation results in the following table.
>
> | Methods | ES | TE | Avg. |
> |:--|:--:|:--:|:--:|
> | Qwen2.5-0.5B | 36.64 | 25.69 | 31.17 |
> | + CPT$_{L \\& A}$ | 32.90 | 22.43 | 27.67 |
> | + CPT$_{A}$ | 32.62 | 25.26 | 28.94 |
> | + MAET w/o API | 36.72 | 28.91 | 32.82 |
> | + MAET | **36.91** | **29.62** | **33.27** |
>
> | Methods | ES | TE | Avg. |
> |:--|:--:|:--:|:--:|
> | Gemma2-2B | 43.41 | 30.01 | 36.71 |
> | + F-CPT$_{L \\& A}$ | 38.48 | **30.39** | 34.62 |
> | + F-CPT$_{A}$ | 37.83 | 25.39 | 31.61 |
> | + MAET w/o API | 43.23 | 29.59 | 36.41 |
> | + MAET | **43.62** | 30.37 | **37.00** |
>
> By comparing the performance of MAET and the baseline methods, we can observe that MAET can also enhance the performance of small-scale models and outperform competitive baselines. Therefore, the evaluation results have shown the effectiveness of MAET and verified that MAET is a general LLM enhancement technology.
>
> > The choice of some hyperparameters is not clear, such as \mu in equation (3).
>
> Actually, in our paper, we have provided the important hyper-parameters in Table 5, and hope the information can help researchers better understand our work and conduct an in-depth study. To better help the reproduction and research, we provide the whole hyper-parameters in the following tables.
>
> **Extraction Stage**
> | Hyper-parameters | Value |
> |:--:|:--:|
> |Learning Rate | $5\times10^{-5}$ |
> | Batch Size | 1M Tokens |
> | Training Steps | 2B Tokens |
> | $\alpha$ in Extraction | 0.8 |
> | $\beta$ in Extraction | 0.2 |
> | Ratio of Key Neurons | 5\% |
>
>
> **Transferring Stage**
> | Hyper-parameters | Value |
> |:--:|:--:|
> |Learning Rate | $5\times10^{-5}$ |
> | Batch Size | 1M Tokens |
> | Training Steps | 2B Tokens |
> | $\gamma$ in Transfer | 0.2 |
> | $\eta$ in Transfer | 1.0 |
> | Ratio of Key Tensors | 80\% / 60\% |
> | $\mu$ for Spanish | 1.5 |
> | $\mu$ for Chinese | 2.0 |
> | $\mu$ for Bengali | 1.2 |
> | $\mu$ for Telugu | 1.2 |
>
> > This paper involves many abbreviations. It is recommended to bold the first appearance of the baseline and dataset names in Section 5.1 for clarity.
> >
> > The related work on model merging is missing key citations [1][2], which demonstrate significantly better performance on multilingual mathematical tasks (MGSM dataset). The authors should reference these studies and engage with their findings in the discussion.
>
> Thanks the suggestions from the reviewer. We will clearly clarify the baseline and dataset names, and add the related studies in our discuccion in our next revision.

---

> ### Author Response · Authors · 2024-11-22
> **Response to the Question**
>
> > Is the “Ability-related Parameter Tensor Selection” in Section 4.2 more closely related to the extraction stage in Section 4.1 rather than the transfer stage?
>
> In Section 4.1, we discuss the method to obtain the transferable ability-related weights to decompose and extract the abilities of LLM. In Section 4.2, we introduce the approach to transfer the abilities by leveraging the ability-related weights obtained in the first stage. In the first step of the transferring stage, i.e., ''Ability-related Parameter Tensor Selection'' , we consider selecting the tensors that are related to the specific ability and irrelevant to the language, and only transfer these ability-related tensors to build the final LLM. In other words, in the extraction stage, we focus on extracting the weights related to the target ability, which does not consider the influence of language in the downstream scenarios. While in the transferring stage, we focus on transferring the language-agnostic weights based on the downstream multi-lingual scenarios and transfer the ability from single-lingual scenarios to multi-lingual scenarios.
> In conclusion, ''Ability-related Parameter Tensor Selection'' is the step to select the important tensor from ability-related weights to perform transferring, which is more closely related to the transfer stage.
>
> > The concepts of “tensor selection” and “parameter weights” are easily confused. Can add examples or annotate the dimensions to distinguish them?
>
> In our paper, ''ability-related tensor'' and ''ability-related weights'' both denote to the parameters (i.e., matrixes of each model layer) of the large language model, while they focus on different aspects. The concept of ''ability-related tensor'' focuses on whether this parameter is the key parameter or is related to the corresponding ability, and the concept of "parameter weights" focuses on the specific value of the LLM parameters. In practice, ''ability-related tensor'' is utilized to identify which parameter should be transferred and ''ability-related weights'' are utilized to calculate the value of the parameter of the final LLM.

---

> ### Author Response · Authors · 2024-11-28
> **Kindly Reminder**
>
> Dear Reviewer 4Vk6,
>
> Thanks for your careful reading of our paper. We carefully explained your concerns and tried our best to elaborate the unclear points. We would like to know whether you find our response satisfactory, or if there are more questions that we could clarify. We are more than happy to hear your comments and address any of your further concerns during the remaining time.
>
> Best,
>
> Authors

---

> > ### Comment · Reviewer_4Vk6 · 2024-11-30
> > **Response to Authors**
> >
> > Thank you for the additional experiments. I have carefully read your response, the updated paper, and the discussions with other reviewers. Based on this, I decided to increase my Presentation and Contribution scores from 2 to 3.
> >
> > I still have some concerns regarding the effect of MAET. The improvement on Gemma2-2B appears to be somewhat modest. Could the authors provide further clarification on this?
> >
> > Additionally, the performance on the MGSM dataset is notably lower than the current SOTA methods. While I understand it may not be necessary to surpass SOTA, could the authors elaborate on the unique advantages of MAET over SOTA, such as reduced training costs?

---

> > > ### Author Response · Authors · 2024-11-30
> > > **Appreciate for raising score And Response for the reviewer's concerns**
> > >
> > > Thank you so much for raising the scores of Presentation and Contribution. We would greatly appreciate the opportunity to engage in discussions with you. The following is our response to your concers, and we hope these responses can address your concerns about our paper.
> > >
> > > > I still have some concerns regarding the effect of MAET. The improvement on Gemma2-2B appears to be somewhat modest. Could the authors provide further clarification on this?
> > >
> > > We present the evaluation results of Qwen2.5-1.5B and Gemma2-2B in the following.
> > >
> > > | Methods | ES | TE | Avg. |
> > > |:--|:--:|:--:|:--:|
> > > | Qwen2.5-0.5B | 36.64 | 25.69 | 31.17 |
> > > | + CPT$_{L \\& A}$ | 32.90 | 22.43 | 27.67 |
> > > | + CPT$_{A}$ | 32.62 | 25.26 | 28.94 |
> > > | + MAET w/o API | 36.72 | 28.91 | 32.82 |
> > > | + MAET | **36.91** | **29.62** | **33.27** |
> > >
> > > | Methods | ES | TE | Avg. |
> > > |:--|:--:|:--:|:--:|
> > > | Gemma2-2B | 43.41 | 30.01 | 36.71 |
> > > | + F-CPT$_{L \\& A}$ | 38.48 | **30.39** | 34.62 |
> > > | + F-CPT$_{A}$ | 37.83 | 25.39 | 31.61 |
> > > | + MAET w/o API | 43.23 | 29.59 | 36.41 |
> > > | + MAET | **43.62** | 30.37 | **37.00** |
> > >
> > > Based on the evaluation results, we observe that training on the ability-related corpus (i.e., $F-CPT_{A}$) does not outperform training on the combined multilingual and ability-related corpus (i.e., $F-CPT_{L\\&A}$). This suggests that the ability-related corpus may not be suitable for Gemma2-2B and does not enhance its performance. In this case, CPT-based methods fail to effectively adapt LLMs to the target scenarios. In contrast, our MAET approach extracts ability-related weights from the model trained on the ability-related corpus and applies these weights to the backbone model. This process helps mitigate the negative effects of directly training the backbone model on the ability-related corpus, resulting in a more effective final model.
> > >
> > > > Additionally, the performance on the MGSM dataset is notably lower than the current SOTA methods. While I understand it may not be necessary to surpass SOTA, could the authors elaborate on the unique advantages of MAET over SOTA, such as reduced training costs?
> > >
> > > For MSGM tasks and mathematical reasoning tasks, current state-of-the-art (SOTA) methods [1][2][3] often focus on the **SFT process** and extensively use powerful models (such as GPT-4) to **synthesize problems** and train LLMs on large-scale, high-quality instruction datasets. The main strategy of these methods involves leveraging an ability-related corpus to train the LLM, similar to the CPT-based methods evaluated in our study. However, according to our evaluation results, our approach, MAET, outperforms the CPT-based baselines, positioning it as a more effective solution and a potential advancement over the current SOTA approaches.
> > >
> > > In our work, we focus on the **pre-training stage** of the LLM training process and use the **same backbone model** (i.e., LLaMA 3.1 8B) and **same training corpus** (i.e., OpenWebMath) for both MAET and other baselines to ensure a fair comparison. Since we do not incorporate synthesized training data or other instruction datasets in this stage, our results differ from those of current SOTA methods. However, these instruction datasets could also be integrated into the MAET framework during the subsequent SFT process, potentially enhancing the LLM's capabilities. In this work, we concentrate on the pre-training stage and leave the analysis of the SFT stage for future investigation.
> > >
> > > In addition to achieving better performance, our MAET significantly reduces the training cost when adapting LLMs to new multilingual scenarios. Specifically, current SOTA methods (e.g., CPT-based methods) require training LLMs on a mixture of multilingual and ability-related corpora. In contrast, our MAET trains LLMs solely on the ability-related corpus and uses simple operations to obtain the final model. The comparison demonstrates that MAET requires substantially fewer computational resources than current SOTA methods, highlighting the efficiency of our approach.
> > >
> > > In conclusion, our MAET approach outperforms SOTA methods under the same evaluation settings and requires fewer computational resources to transfer LLMs to a new domain, demonstrating its superior efficiency and effectiveness.
> > >
> > > [1] MAmmoTH2: Scaling Instructions from the Web
> > >
> > > [2] JiuZhang3.0: Efficiently Improving Mathematical Reasoning by Training Small Data Synthesis Models
> > >
> > > [3] Qwen2.5-Math Technical Report: Toward Mathematical Expert Model via Self-Improvement

---

> > > > ### Comment · Reviewer_4Vk6 · 2024-12-01
> > > > **Response to Authors**
> > > >
> > > > Thanks to the authors for the positive response. I am optimistic about the novelty of this paper. However, my rating remains relatively conservative due to concerns regarding the improvement of multilingual tasks in this work.
> > > >
> > > > My concern is whether the authors have fully explored the potential of the continual pre-training methods they compared. I noticed that the authors incorporated translation-based methods. While it is understandable that LLMs may perform less well than MAET in this setting due to translation quality, existing researches suggest that it is not always necessary to translate downstream task data for continual pre-training. For instance, using more general and widely available translation data can also improve the multilingual capabilities of LLMs. Specifically, [1] pointed out that using only the translation data for downstream task queries can significantly enhance multilingual performance, as queries are typically shorter, making the cost and errors of translation more manageable.
> > > >
> > > > In summary, I acknowledge the authors' novelty and the potential value of this work. Considering its effect on multilingual tasks, I maintained my rating.
> > > >
> > > > Good luck with your research!
> > > >
> > > > [1] Question Translation Training for Better Multilingual Reasoning (ACL Findings 2024)

---

> > > > > ### Author Response · Authors · 2024-12-02
> > > > > **Response to Reviewer**
> > > > >
> > > > > Dear reviewer 4Vk6,
> > > > >
> > > > > Thanks for your comment and acknowledgment of our work.
> > > > >
> > > > > Regarding the improvement of our approach, MAET can **significantly enhance** performance in various downstream scenarios, such as mathematical reasoning tasks on the LLaMA3-8B model and scientific reasoning tasks on the Qwen-1.5B model. Although MAET provides only minor improvements in some other scenarios, it still achieves better or comparable performance compared to strong baseline methods (e.g., CPT-based methods), while **requiring very few computational resources**. This demonstrates that MAET is both an **effective** and **efficient** method for adapting LLMs to multilingual tasks, making it especially valuable for real-world applications, particularly in low-resource scenarios.
> > > > >
> > > > > Best,
> > > > >
> > > > > Authors

---

> ### Author Response · Authors · 2024-12-03
> **Kindly Reminder**
>
> Dear reviewer 4Vk6,
>
> Thanks for your careful reading of our paper and the positive score. We carefully explained your concerns and tried our best to elaborate the unclear points. We would like to know whether you find our response satisfactory, or if there are more questions that we could clarify. Since the rebuttal stage is coming to an end, we are more than happy to hear your comments and address any of your further concerns during the remaining time.
>
> Best,
>
> Authors

---

### Official Review · Reviewer_HfAo · 2024-11-11

**Soundness:** 3
**Presentation:** 1
**Contribution:** 2
**Rating:** 3
**Confidence:** 4

**Summary:**

Given that related work mostly trains LLMs with ability-related corpus in target languages, which is relatively rare in low-resource languages, the authors propose MEAT, an approach to multilingual ability extraction and transfer. The basic idea is to identify the "ability-specific weights" inside the LLMs, and decompose them from the "language-specific" ones. By examining the weight changes after a post-pretraining phase, it identifies the top-k neurons as the "key neurons". Then, it trains the key neurons on two different corpora, and further obtains ability-specific weights for the identified key neurons. Finally, it conducts the cross-lingual ability transfer by weighted average among several weights that are trained on various corpora. Experimental results show that MEAT outperforms continual pre-training, domain adaptation and task vector approaches on the multilingual mathematical and scientific tasks.

**Strengths:**

- Multilingual LLM is an important research area for improving the accessibility of LLMs for non-English users. Obviously, current LLMs are not good at processing non-English tasks, especially in low-resource languages. Thus, research on how to transfer abilities or knowledge to low-resource languages is meaningful.
- Identifying and decomposing the "ability-specific" weights of LLMs is an interesting idea, which would potentially further enhance the more types of capabilities of LLMs in low-resource languages.
- Experimental results show that MEAT outperforms continual pre-training, demonstrating its effectiveness on cross-lingual transfer.

**Weaknesses:**

- Although there are few human-labeled multilingual instruction data (aka ability-specific data, or downstream task data), synthetic data generation is currently a practical way towards multilingual LLMs. For example, the translate-train method is a simple and strong baseline for multilingual tasks [1] [2], which simply translates the instruction data into other languages, achieving significant improvements. Another way to synthesize data is to prompt LLMs (e.g., GPT-4) to generate multilingual responses such as Bactrian-x [3]. Besides, the translate-test pipeline is also commonly used in multilingual evaluation (e.g., MEGA [4], XLM-R [5]). Therefore, it is necessary to clarify why we should use MEAT rather than these common approaches.
- Some terms used in the paper are confusing. There are "ability-related weights", "key neurons", "ability-specific weights", "ability-related tensors", and "language-specific weights" mentioned in the paper, and the relation and differences among the terms are not clear. For instance,  what the difference is between "ability-specific weights" and "ability-related weights".  At L236, R(L_1) is a typeof ability-related weights, and R(A_i) is a type of ability-specific weights according to Eq (2). What is the language-specific weight mentioned at L054?
- Minor: In Table 5 (Appendix), both stages are labeled as "Extraction".

[1] ERNIE-M: Enhanced Multilingual Representation by Aligning Cross-lingual Semantics with Monolingual Corpora

[2] ENHANCING CROSS-LINGUAL TRANSFER BY MANIFOLD MIXUP

[3] Bactrian-x: Multilingual replicable instruction-following models with low-rank adaptation

[4] MEGA:Multilingual Evaluation of Generative AI

[5] Unsupervised Cross-lingual Representation Learning at Scale

**Questions:**

Please clarify the relations and differences among "ability-related weights", "key neurons", "ability-specific weights", "ability-related tensors", and "language-specific weights".

---

> ### Author Response · Authors · 2024-11-22
> **Response to the Weakness - Part 1**
>
> We sincerely thank the reviewer for the insightful suggestions.
>
> > Although there are few human-labeled multilingual instruction data (aka ability-specific data, or downstream task data), synthetic data generation is currently a practical way towards multilingual LLMs. For example, the translate-train method is a simple and strong baseline for multilingual tasks [1] [2], which simply translates the instruction data into other languages, achieving significant improvements. Another way to synthesize data is to prompt LLMs (e.g., GPT-4) to generate multilingual responses such as Bactrian-x [3]. Besides, the translate-test pipeline is also commonly used in multilingual evaluation (e.g., MEGA [4], XLM-R [5]). Therefore, it is necessary to clarify why we should use MEAT rather than these common approaches.
>
> Actually, the translation-based methods require more computation resources in the continual pretraining process, due to the heavy use of translation models for data synthesis (e.g., GPT-4o or Claude 3). It causes high expense of translation and is hard to employ in practice. Besides, the errors of the translation model may also be accumulated to hurt the final performance.
>
> Following the suggestions from the reviewer, we conduct the translation-based method (i.e., ''CPT_{T}'') in our evaluation process and present the results on multi-lingual scientific tasks in the following table. For the details about the translation-based method, we first utilize GPT-4o which is one of the most powerful multi-lingual LLMs to translate the domain-specific corpus from English to other languages, including Spanish, Chinese, Bengali, and Telugu, and then leverage the translated corpus to continually pre-train the backbone model.
>
> | Methods | ES | ZH | BN | TE | Avg. |
> |:--:|:--:|:--:|:--:|:--:|:--:|
> | Backbone | 55.06 | 47.24 | 36.63 | 29.26 | 42.05 |
> | + CPT_{T} | 50.35 | 45.36 | 34.54 | **34.46** | 41.18 |
> | + CPT_{S \& T} | 53.73 | 46.30 | 35.06 | 31.73 | 41.71 |
> | + CPT_{S} | 51.90 | 45.71 | 33.35 | 29.41 | 40.09 |
> | + MAET | **56.20** | **48.00** | **37.64** | 30.38 | **43.06** |
>
> According to the experimental results in the above table, we can observe that our MAET outperforms the translation-based method. The translation-based method consumes more computational resources and cannot achieve better performance. The reason might be that the errors in the translated corpus would mislead LLMs and even cause overfitting. In contrast, our approach decomposes the scientific ability and language ability, and transfers the scientific ability from one language to another, preventing overfitting, decreasing the expense, and improving performance.

---

> ### Author Response · Authors · 2024-11-22
> **Response to the Weakness and Question - Part 2**
>
> > Some terms used in the paper are confusing. There are "ability-related weights", "key neurons", "ability-specific weights", "ability-related tensors", and "language-specific weights" mentioned in the paper, and the relation and differences among the terms are not clear. For instance, what is the difference between "ability-specific weights" and "ability-related weights". At L236, R(L_1) is a type of ability-related weights, and R(A_i) is a type of ability-specific weights according to Eq (2). What is the language-specific weight mentioned at L054?
>
> In this work, we focus on building multi-lingual ability-enhanced LLMs and propose a novel approach to achieve effective and efficient extracting and transferring abilities. As our paper introduces many novel techniques, we try to utilize the above terms from existing papers to better explain our approach. To make them more clearly and concise, we clarify the definition and the relation of the concepts mentioned in our paper.
>
> **Key Neurons**. Neuron refers to one of the trainable values of the tensors in LLMs. As previous work pointed out [1], different neurons might control the different abilities of LLMs. Following this finding, in our work, we define the neurons that control the specific ability as the ''Key Neurons''. Key neurons can be regarded as a set without duplication, and a neuron belonging to the set means that this neuron can control the specific ability [2]. During the following training process, only the neurons belonging to the key neurons will be trained and optimized.
>
> **Ability-related Weights**. Ability-related weights refer to the value of the whole neuron in LLM, which can represent the corresponding ability of LLM [3][4]. In MAET, we obtain the ability-related weights through equation 2. The ability-related weights contain the value of all neurons. Since only the key neurons will be trained during the training process, the value of the neurons not belonging to key neurons is zero in the ability-related weights.
>
> **Ability-specific Weights**. In our work, the definition of ability-specific weights is similar to the definition of ability-related weights. The meaning of ability-specific weights refers to the value of the key neurons. In our next revision, we will unify the expression of ''ability-related weights'' and ''ability-specific weights'' into ''ability-related weights'', to reduce ambiguity of our paper and concepts.
>
> **Ability-related Tensors**. Ability-related tensors can be regarded as a set of LLM tensors, which is high-related to the corresponding ability. Previous work has studied how the LLM layers influence the ability [5]. Different from key neurons, ability-related tensors focus on the higher-level information, integrating the sparse neurons into a coarser-grained element [6]. A tensor belonging to the ability-related tensors denotes that this tensor is high-related to the corresponding ability and can control this ability.
>
> **Language-specific weights**. Similar with the ability-related weights, language-specific weights also refer to the value of the whole neurons in LLMs [7]. However, language-specific weights represent the language abilities of LLM that include multiple abilities (i.e., one language can be regarded as one ability) [8], and the method of obtaining them is also different from ability-specific weights. In MAET, we first calculate the ability-related weights of each language and then Integrating these weights together to obtain the language-specific.
>
> In conclusion, ''Key Neurons'' and ''Ability-related Tensors'' denote LLM inner neurons and tensors,which are related to the corresponding abilities. And ''Ability-related Weights'' and ''Language-specific weights'' denote the value of the whole neurons of LLMs, which can control the abilities or languages.
>
> [1] Let’s Focus on Neuron: Neuron-Level Supervised Fine-tuning for Large Language Model
>
> [2] Not Everything is All You Need: Toward Low-Redundant Optimization for Large Language Model Alignment
>
> [3] Language Models are Super Mario: Absorbing Abilities from Homologous Models as a Free Lunch
>
> [4] Editing Models with Task Arithmetic
>
> [5] Enhancing Translation Ability of Large Language Models by Leveraging Task-Related Layers
>
> [6] Configurable Foundation Models: Building LLMs from a Modular Perspective
>
> [7] Unveiling Linguistic Regions in Large Language Models
>
> [8] Language-specific neurons: The key to multilingual capabilities in large language models
>
> > Minor: In Table 5 (Appendix), both stages are labeled as "Extraction".
>
> Thanks for the suggestion from the reviewer. We will correct the typos in the new revision of our paper.

---

> ### Author Response · Authors · 2024-11-28
> **Kindly Reminder**
>
> Dear Reviewer HfAo,
>
> Thanks for your careful reading of our paper. We carefully explained your concerns and tried our best to elaborate the unclear points. We would like to know whether you find our response satisfactory, or if there are more questions that we could clarify. We are more than happy to hear your comments and address any of your further concerns during the remaining time.
>
> Best,
>
> Authors

---

> ### Author Response · Authors · 2024-12-02
> **Kindly Reminder and Summary of Author Response**
>
> Dear Reviewer HfAo,
>
> Thanks for your careful reading of our paper. Since the rebuttal stage is coming to an end, we are more than happy to hear your comments and address any of your further concerns during the remaining time.
>
> Below is the summary of our response:
>
> + For translation-based methods, while they can construct a multilingual ability-related corpus, they require a powerful multilingual model for the translation process, which demands **significant computational resources**, and the **quality of the generated corpus may be low**. In contrast, our approach, MAET, only requires a **single-lingual ability-related corpus** and **fewer computational resources**, yet it can effectively and efficiently adapt LLMs to multilingual scenarios.
>
> + In our paper, we have carefully defined and explained the key concepts, and provided pseudo code to help readers better understand our approach. Since our paper introduces several novel techniques, we have made an effort to use a combination of simple, well-established terms from previous work to explain these concepts. We believe that our paper is easy to understand and follow.
>
> Hope our responses and explanations can address your concerns.
>
> Best,
>
> Authors

---

> ### Author Response · Authors · 2024-12-03
> **Kindly Reminder**
>
> Dear reviewer HfAo,
>
> Thanks for your careful reading of our paper and the positive score. We carefully explained your concerns and tried our best to elaborate the unclear points. We would like to know whether you find our response satisfactory, or if there are more questions that we could clarify. Since the rebuttal stage is coming to an end, we are more than happy to hear your comments and address any of your further concerns during the remaining time.
>
> Best,
>
> Authors

---

### Author Response · Authors · 2024-11-25
**General Response**

Thanks to the insightful suggestions from reviewers. Below, we summarize our responses to major concerns, and upload a new revision of our paper.

**Key Concepts in Our Paper**

In our paper, there are four key concepts, i.e., Key Neurons, Ability-related Tensors, Ability-related Weights, and Language-specific weights. ''Key Neurons'' and ''Ability-related Tensors'' denote LLM inner neurons and tensors, which are related to the corresponding abilities. And ''Ability-related Weights'' and ''Language-specific weights'' denote the value of the whole neurons of LLMs, which can control the abilities or languages.

**Experiment about the Translation-based Baselines**

Translation-based methods require large computation resources, which is difficult to be employed in the pre-training stages. To conduct more comprehensive experiments, we adopt this kind of method into our evaluation and present the results in the following table. We can observe that our MAET also outperforms the translation-based methods and only needs less expenses.

| Methods | ES | ZH | BN | TE | Avg. |
|:--:|:--:|:--:|:--:|:--:|:--:|
| Backbone | 55.06 | 47.24 | 36.63 | 29.26 | 42.05 |
| + CPT_{T} | 50.35 | 45.36 | 34.54 | **34.46** | 41.18 |
| + CPT_{S \& T} | 53.73 | 46.30 | 35.06 | 31.73 | 41.71 |
| + CPT_{S} | 51.90 | 45.71 | 33.35 | 29.41 | 40.09 |
| + MAET | **56.20** | **48.00** | **37.64** | 30.38 | **43.06** |

**Experiment about Different Backbone Models**

To further assess the effectiveness and practicality of our approach, we employ MAET on different backbone models and evaluate the performance in multi-lingual scenarios. The evaluation results have verified that our approach MAET is a general LLM enhancement technology.

| Methods | ES | TE | Avg. |
|:--|:--:|:--:|:--:|
| Qwen2.5-0.5B | 36.64 | 25.69 | 31.17 |
| + CPT$_{L \\& A}$ | 32.90 | 22.43 | 27.67 |
| + CPT$_{A}$ | 32.62 | 25.26 | 28.94 |
| + MAET w/o API | 36.72 | 28.91 | 32.82 |
| + MAET | **36.91** | **29.62** | **33.27** |

| Methods | ES | TE | Avg. |
|:--|:--:|:--:|:--:|
| Gemma2-2B | 43.41 | 30.01 | 36.71 |
| + F-CPT$_{L \\& A}$ | 38.48 | **30.39** | 34.62 |
| + F-CPT$_{A}$ | 37.83 | 25.39 | 31.61 |
| + MAET w/o API | 43.23 | 29.59 | 36.41 |
| + MAET | **43.62** | 30.37 | **37.00** |

**Details about the Revision**

We have revised our paper and uploaded the new revision, where we use blue color to identify the modification. In this revision, the major updates are as follows,
+ Adding the experiments on more backbone LLMs (i.e., Qwen2.5 and Gemma2) to further validate the effectiveness and practicality of our method.
+ Adding the translation-based methods into the main results and the analysis of these new baselines in section 5.2.
+ Adding the key citations into the ''Model Merging'' part of related work.
+ Clearly clarifying the concepts of our paper, including ''Key Neurons'', ''Ability-related Weights'', ''Ability-related Tensors'', and ''Language-specific Weights''.
+ Emphasizing the first appearance of the baselines and dataset.
+ Correcting the typos in Table 5 and Algorithm 1.

---

### Note · Authors · 2024-12-13

I have read and agree with the venue's withdrawal policy on behalf of myself and my co-authors.